# Position: Predictive Uncertainty Is Not Enough — Joint Distribution for Full Uncertainty Representation

**Adrià Aldomà** [* 1 2]  **Unai Gurbindo** [* 1 2]  **Axel Brando** [1]

## Abstract

When AI is deployed in safety-critical domains, erroneous and overconfident predictions can have severe consequences, making comprehensive uncertainty quantification (UQ) essential for responsible decision-making. Current UQ methods based on epistemic and aleatoric decomposition are insufficient for fully understanding the problem, a limitation further compounded by treating these terms in isolation from uncertainty about the domain. Our position claims that any meaningful analysis must account for three sources of uncertainty — domain, epistemic, and aleatoric — and that only the joint distribution $p(x, y|\mathcal{D})$ provides a coherent representation of them. We mirror prior findings showing that information-theoretic UQ methods are suboptimal in ID and OOD settings, mainly due to the difficulty of disentangling epistemic and aleatoric components. We argue that modeling the unconditional distribution $p(x|\mathcal{D})$ is necessary to account for input validity, giving rise to a third class of uncertainty: *domain* uncertainty. Finally, by jointly considering the domain distribution and the conditional distribution $p(y|x, \mathcal{D})$, we argue that their product, $p(x, y|\mathcal{D})$, captures all relevant sources of uncertainty.

## 1. Introduction

As Machine Learning (ML) systems are increasingly deployed in high-stakes settings, *safety* and *trustworthiness* hinge on more than raw predictive accuracy. The actual ethical and regulatory situation demands systems to recognize when to abstain or defer, and when the operating

---
*Equal contribution [1]Barcelona Supercomputing Center (BSC), Barcelona, Spain [2]Universitat Politècnica de Catalunya (UPC), Barcelona, Spain. Correspondence to: Adrià Aldomà <adria.aldoma@upc.edu>, Unai Gurbindo <unai.gurbindo@upc.edu>.

*Proceedings of the 43$^{rd}$ International Conference on Machine Learning*, Seoul, South Korea. PMLR 306, 2026. Copyright 2026 by the author(s).

conditions no longer match those assumed during development and thus their outputs are unreliable, with effective *human oversight* as required by Article 14 of the EU AI Act (European Parliament & Council of the European Union, 2024). Overconfident failures under ambiguous inputs, rare events or distribution shift motivate uncertainty-aware decision rules as a practical safety mechanism, rather than a mere evaluation add-on.

Uncertainty Quantification (UQ) has therefore become central in ML, as it provides a principled way to assess the reliability of model outputs and to support safe behaviors such as rejection, fallback, or human handover. Traditionally, uncertainty is decomposed into two components: *aleatoric* and *epistemic* (Kiureghian & Ditlevsen, 2009; Kendall & Gal, 2017). This distinction is often motivated by their different reducibility properties. Aleatoric uncertainty (AU) captures the inherent noise, ambiguity, or occlusion in the data and is therefore considered irreducible, while epistemic uncertainty (EU) reflects limitations in the model or training data and is, in principle, reducible through additional data or improved modeling.

From a trustworthiness perspective, such a decomposition promises interpretability and actionable insight into the sources of uncertainty. However, recent work has increasingly questioned the clarity of the validity and practical usefulness of this classical dichotomy. Information-theoretic analyses and empirical evaluations have shown that disentangling predictive entropy into aleatoric and epistemic components is far less straightforward than commonly assumed, particularly in realistic, high-dimensional settings (Mucsányi et al., 2024; de Jong et al., 2026).

Not only but more importantly, when the data-generating process shifts, the predictive entropy limitation is further compounded by the systematic isolation of epistemic and aleatoric from uncertainty about the *domain* itself. In practice, domain-related mismatches such as anomalies or outliers, covariate shift and semantic shift, and out-of-distribution in general is often treated heuristically as a separate detection problem–for which multiple approaches have been proposed (Yang et al., 2024a)–rather than being integrated into a unified uncertainty framework as proposed in (Brando et al., 2023).

In this position paper, we argue that **any meaningful uncertainty analysis must explicitly account for three sources of uncertainty — domain, epistemic, and aleatoric — and that only the joint distribution $p(x, y|\mathcal{D})$ provides a coherent and complete representation of uncertainty**. From this perspective, modeling the product of the unconditional $p(x|\mathcal{D})$ and conditional $p(y|x, \mathcal{D})$ distribution is essential to assess input validity and within-domain uncertainty.

The structure of this position paper is divided as follows. **Section 2** formalizes the uncertainty estimation problem and reviews prior work on the dichotomy of epistemic and aleatoric uncertainty, highlighting conceptual tensions and limitations. In **Section 3** we derive theoretical bounds on the expressive capacity of predictive uncertainty and argue that aleatoric and epistemic uncertainty are neither cleanly separable nor sufficient to address domain-level failures, since domain uncertainty is itself a first-class source of uncertainty. Then, we defend arguments for modeling the domain distribution and finally propose a joint treatment that clarifies how uncertainty sources interact rather than assuming clean separability.[1] Finally, **Section 4** presents alternative views to our position.

## 2. Motivation

### 2.1. Predictive Distribution and Uncertainty

We assume a supervised learning setting with a probabilistic predictor (e.g., a neural network) trained on a dataset $\mathcal{X}$ and $\mathcal{Y}$, where $X \in \mathcal{X}$ and $Y \in \mathcal{Y}$ denote input and output random variables. Let the training data be drawn from their joint distribution, $\mathcal{D} = \{(x_i, y_i)\}_{i=1}^{N} \sim p(X, Y)$, where $x_i \in \mathcal{X}$ and $y_i \in \mathcal{Y}$ are realizations. For a given input $x$, the data-generating process induces a conditional distribution $p(Y|X = x)$, where $p(Y = y|X = x)$ is the probability of observing outcome $y$ given $x$.

To model $p(Y|X)$, we train a parametric model with parameters $\theta \in \Theta$ that, given a test input $x$, induces a predictive distribution over $Y$. In the Bayesian perspective, uncertainty about $\theta$ is captured by a posterior $q(\theta|\mathcal{D})$, leading to the conditional *predictive distribution*

$$p(y|x, \mathcal{D}) = \int_{\Theta} q(\theta|x, \mathcal{D}) \, p(y|x, \theta) \, d\theta, \quad (1)$$

where $\Theta$ is the space of possible parameter values for the chosen architecture.

For classification problems the output space corresponds to a set of different classes $\mathcal{Y} = \{1, \ldots, K\}$, while regression problems allow the outcome to take any real value.

---

[1] Code for experiments is available at https://github.com/unai-gurbindo/Position-Joint-Distribution-UQ.git

Shannon entropy is the commonly adopted measure of *predictive* uncertainty in such conditional distributions (Houlsby et al., 2011; Gal & Ghahramani, 2016; Depeweg et al., 2018; Hüllermeier & Waegeman, 2021). Information-theoretic identities permit an additive decomposition of the total predictive uncertainty (Depeweg et al., 2018) regardless of the approximation:

$$\mathbb{H}[Y|x, \mathcal{D}] = \mathbb{E}_{q(\theta|x, \mathcal{D})}\big[\mathbb{H}[Y|x, \theta, \mathcal{D}]\big] + \mathbb{I}[Y; \theta|x, \mathcal{D}]. \quad (2)$$

A derivation of this result can be found in Appendix A. The first term, the expected conditional entropy under the posterior, is commonly called *aleatoric* uncertainty; the second, the mutual information (MI) between labels and parameters, is called *epistemic* uncertainty.

Epistemic uncertainty is interpreted as ignorance arising from limited data and imperfect knowledge of the model parameters, whereas aleatoric uncertainty reflects inherent stochasticity in the data generating process, i.e. data ambiguity. However, from an information-theoretic perspective, Equation (2) assumes the model family can represent the true data process, and that sampling uncertainty from $\mathcal{D}$ and model misspecification do not introduce additional sources of uncertainty (Hüllermeier & Waegeman, 2021). These assumptions are contradicted in the following section, so this perspective is seeming to fall apart.

### 2.2. Dichotomy and Contradictions

Uncertainty estimation methods are commonly grouped into two broad families: *distributional methods* and *deterministic methods*.

Distributional methods explicitly represent uncertainty through second-order probabilistic models, such as distributions over predictive categorical distributions (e.g., evidential methods (Sensoy et al., 2018; Charpentier et al., 2020) and their extensions), or through approximate posterior sampling techniques, including Deep Ensembles (Lakshminarayanan et al., 2017), Monte Carlo Dropout (Gal & Ghahramani, 2016), and Bayesian Neural Networks (Neal, 1996).

Deterministic methods, by contrast, output a single scalar uncertainty score $u(x) \in \mathbb{R}$ for each input. These approaches rely on explicit functional definitions of uncertainty, for instance via feature-space density surrogates as in Deep Deterministic Uncertainty (DDU) (Mukhoti et al., 2023) or Deterministic Uncertainty Quantification (DUQ) (Van Amersfoort et al., 2020).

As highlighted by Kirchhof et al. (2025b), the literature exhibits substantial variation in how uncertainty—particularly epistemic uncertainty—is defined and operationalized. Epistemic uncertainty has been variously characterized as the

number of plausible models (Wimmer et al., 2023), disagreement among models (Gal et al., 2017), or low feature-space density (Mukhoti et al., 2023; Van Amersfoort et al., 2020). Aleatoric uncertainty, in turn, has been associated with the behavior of a Bayes-optimal predictor (Schweighofer et al., 2025; Bengs et al., 2022), with pointwise ground-truth variance (Lahlou et al., 2023) or more generally with the conditional distribution of the target variable (Brando et al., 2019).

For distributional approaches grounded in information-theoretic decompositions, definitions of aleatoric uncertainty tend to be more widely agreed upon. As noted by Van Amersfoort et al. (2020), "the only uncertainty that can reliably be captured by looking at the entropy of the softmax distribution is aleatoric uncertainty." Epistemic uncertainty, by contrast, is often treated residually—as "everything else" (Jiménez et al., 2026)—leading to a proliferation of competing interpretations (Hüllermeier & Waegeman, 2021). Taken together, this body of work points to a deeper issue: uncertainty concepts are frequently redefined implicitly and inconsistently across studies, resulting in subtle—and sometimes contradictory—uses of the same terminology (Kirchhof et al., 2025a; Bickford Smith et al., 2025).

Recent studies increasingly challenge whether the classical aleatoric/epistemic dichotomy provides a complete or coherent decomposition of total uncertainty for either distributional or deterministic approaches (Juergens et al., 2024; Jiménez et al., 2026). In the benchmark by Mucsányi et al. (2024), the two components are highly correlated, and in many cases appear functionally interchangeable–echoing observations in (Valdenegro-Toro & Mori, 2022; de Jong et al., 2026). Bickford Smith et al. (2025) argue that the inconsistency originates from foundational issues resulting in nonequivalent mathematical quantities and propose an alternative view from the decision-theoretic perspective. Wimmer et al. (2023) further question whether an additive decomposition is conceptually justified as well as the suitability of MI as EU measure, and Mukhoti et al. (2021; 2023) similarly highlight that the split is not trivial. Others like Schweighofer et al. (2023) propose novel measures of predictive uncertainty that do not necessarily assume that the Bayesian model average (BMA) predictive distribution is equivalent to the true model's predictive distribution and identify the epistemic uncertainty as the expected pairwise KL-divergence proposed by (Malinin, 2019; Malinin & Gales, 2021). As an alternative, Osband et al. (2023) argue that predictions collapse different sources of uncertainty into a single distribution and propose Epistemic Neural Networks to model joint predictions across inputs, addressing this loss of information by enriching the output representation.

All these contradictions in concepts become even more pronounced when we move to scenarios involving the training domain and out-of-distribution instances, which are typically addressed through epistemic uncertainty measures (Kirsch et al., 2021). While this makes sense from the reducibility point of view, it misses the precise nature of what an out-of-distribution sample is. Precisely, in the results of Figure D.1 (Appendix D) of (Mucsányi et al., 2024) authors observe that, for information theory decomposition methods, aleatoric aggregators are better than epistemic ones for OOD. Same observations are found in (Schweighofer et al., 2025) and Appendix A of (Kotelevskii et al., 2025), which is contrary to the general belief that OOD data should show high epistemic uncertainty. Additionally, Ovadia et al. (2019) explore the utility of predictive uncertainty as a score for distribution shift and show bad results for all methods, being a small (size 5) Deep Ensemble the best performing one. Also, surprisingly, simply using Maximum Softmax Probability (MSP) (Hendrycks & Gimpel, 2017) yields acceptable results given the simplicity of the method.

Other OOD detection approaches do not explicitly rely on uncertainty quantification. A popular strategy is the one of (Lee et al., 2018), where a class conditional Mahalanobis distance is computed in the feature space of the pre-trained classifier and used as an OOD score. Hendrycks et al. (2019) propose Outlier Exposure, a training-time regularization strategy in which the classifier is exposed to auxiliary out-of-distribution samples and encouraged to produce uniform predictive distributions on such inputs. Fort et al. (2021) extend this paradigm to low-data regimes, demonstrating that even few-shot outlier exposure can improve OOD detection when fine-tuning pretrained models. Complementary lines of work focus on, for example, applying small input perturbations computed from output gradients to make the model's softmax scores more separable between ID and OOD points (Liang et al., 2018) or using energy-based scores derived from model logits (Liu et al., 2020).

Recent conceptual efforts, such as ImageNet-OOD (Yang et al., 2024b) and (Tian et al., 2021) advocate for a unified conceptualization of dataset shift in OOD detection rather than treating "out-of-distribution" as a single, monolithic phenomenon. Broadly, they define distribution shift as any difference between the training and test joint distributions ($p_{train}(x, y) \neq p_{test}(x, y)$), and categorize *covariate shift* and *semantic (concept) shift* as changes in the input while preserving label and changes in the conditional label distribution, respectively. Li et al. (2025) further underline the need for unified distribution-shift frameworks by showing that many existing OOD detection methods are fundamentally misaligned with meaningful definitions of distributional drift, conflating semantic vs. covariate changes and mistaking uncertainty or feature distance for true distributional divergence.

In general, the reviewed literature suggests that the epis-

temic/aleatoric division does not produce stable and operationally consistent signals, especially in the case of distribution change, as predictive uncertainty confuses ambiguity within the domain with uncertainty about whether an input belongs to the domain.

## 3. Towards a Complete Representation of Uncertainty

### 3.1. Limitations of Information Theoretic Predictive Uncertainty

As explained before, the dominant paradigm in uncertainty quantification treats predictive uncertainty as decomposable into independent aleatoric and epistemic components via the information-theoretic identity in Equation (2). In this subsection, we argue that such an interpretation is problematic in practical ensemble-based settings and, crucially, insufficient for safety-oriented tasks such as OOD detection. Our central claim is that predictive uncertainty is structurally bounded by the ensemble mixture itself and therefore cannot reliably express distributional novelty. This motivates the explicit modeling of domain uncertainty, formalized as $p(x|\mathcal{D})$, as a separate and indispensable component of uncertainty quantification.

To understand the limitations of the aforementioned paradigm, we analyze the structure of predictive entropy via finite ensembles. In the Bayesian setting, the posterior predictive distribution is approximated via the BMA (Wilson & Izmailov, 2020):

$$p(y|x, \mathcal{D}) \approx \sum_{m=1}^{M} p(\theta_m|x, \mathcal{D}) \, p(y|x, \theta_m, \mathcal{D}), \quad (3)$$

where $\{\theta_m\}_{m=1}^{M}$ are samples or approximations from the posterior over model parameters. For any finite $M$, the predictive distribution is a finite mixture, and its entropy necessarily inherits structural properties from this mixture form.

Using the LogSumExp (LSE) identity (see Appendix C), the predictive entropy can be bounded as

$$- \mathbb{E}_{p(y|x)} \left[ \max_m \{ \log \left( w_m p_m(y|x) \right) \}_{m=1}^{M} \right] - \log M$$
$$\leq \mathbb{H}[Y|x, \mathcal{D}] \leq \quad (4)$$
$$- \mathbb{E}_{p(y|x)} \left[ \max_m \{ \log \left( w_m p_m(y|x) \right) \}_{m=1}^{M} \right].$$

where $w_m = p(\theta_m|x, \mathcal{D})$ and $p_m(y|x) = p(y|x, \theta_m, \mathcal{D})$. These bounds reveal that predictive entropy is controlled by the upper envelope of the ensemble predictions. For each label $y$, the dominant term is the largest weighted probability $w_m p_m(y|x)$ assigned by any ensemble component,

and the entropy bound averages this quantity over labels according to $p(y|x)$. The remaining mixture contribution is constrained by an additive $\log M$ term.

For uniform mixtures such as Deep Ensembles ($w_m = \frac{1}{M}$), the bounds simplify to:

$$- \mathbb{E}_{p(y|x, \mathcal{D})} \left[ \max_m \log p(y|x, \theta_m, \mathcal{D}) \right]$$
$$\leq \mathbb{H}[Y|x, \mathcal{D}] \leq \quad (5)$$
$$- \mathbb{E}_{p(y|x, \mathcal{D})} \left[ \max_m \log p(y|x, \theta_m, \mathcal{D}) \right] + \log M.$$

Adding an additional model ($m' \in \{1, \ldots, M+1\}$) can only increase the maximal log-likelihood term,

$$\max_m \log p(y|x, \theta_m, \mathcal{D}) \leq \max_{m'} \log p(y|x, \theta_{m'}, \mathcal{D}) \quad (6)$$

but the resulting gain in predictive entropy is strictly limited by the mixture size. This improvement is therefore not controlled by mutual information alone.

When $M = 1$, the bounds collapse to the negative log-likelihood of the single model; as $M$ grows, the width of the interval increases monotonically, but remains strictly limited by $\log M$. No matter what the input is, the gap between the "hard" (single-component) and "soft" (mixture-based) views is at most $\log M$. Consequently, epistemic uncertainty derived from mixture disagreement is fundamentally capacity-limited.

**Visualizing entropy bounds and saturation effects.** Figure 1 visualizes these bounds for a 5 model ensemble trained on CIFAR10 (Krizhevsky & Hinton, 2009) and evaluated on a mixed ID/OOD test set with SVHN images (Netzer et al., 2011). The predictive entropy consistently lies within the theoretical bounds, confirming that uncertainty values are tightly constrained by ensemble size. As the lower bound increases, aleatoric uncertainty rises, and epistemic uncertainty follows proportionally (see Figure 2).

While OOD samples (SVHN) in Figure 1 concentrate toward higher predictive entropy regions, they do not exhibit systematically higher epistemic uncertainty than in-distribution samples. Instead, high-entropy regions contain a mixture of ID and OOD points. As uncertainty increases, the fraction of OOD samples also increases, so OOD samples are concentrated high-entropy regions, but predictive entropy still does not identify them point-by-point. This illustrates a critical limitation of uncertainty-based OOD detection: OOD inputs are filtered only by co-locating with many valid ID samples, leading to high false-positive rejection. This is a prevalence effect, not a discriminative one. Therefore, apparent OOD sensitivity emerges not because epistemic uncertainty uniquely captures distributional shift, but because both OOD and uncertain ID points accumulate near the upper entropy bound.

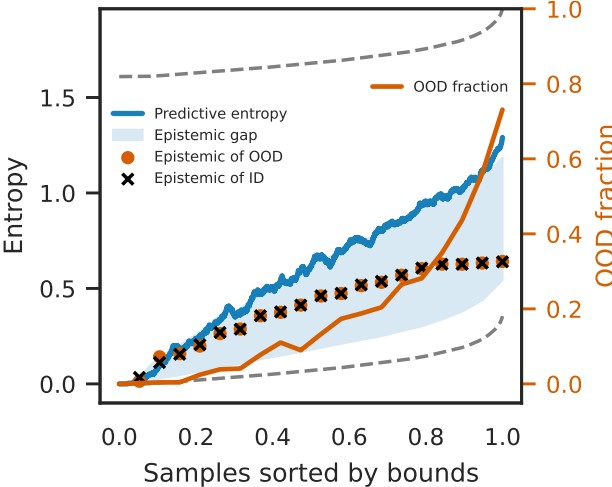

*Figure 1.* Predictive entropy, entropy bounds, and out-of-distribution (OOD) concentration for a 5-model ensemble trained on CIFAR-10 and evaluated on a mixed test set consisting of 80% CIFAR-10 (ID) and 20% SVHN (OOD) samples. Test **samples are sorted in ascending order according to the bounds**. The predictive entropy consistently lies between the lower and upper bounds given by Equation (5). Epistemic gap fills the space between AU and predictive entropy, i.e., EU. Epistemic uncertainty values for only ID and only OOD points are also plotted.

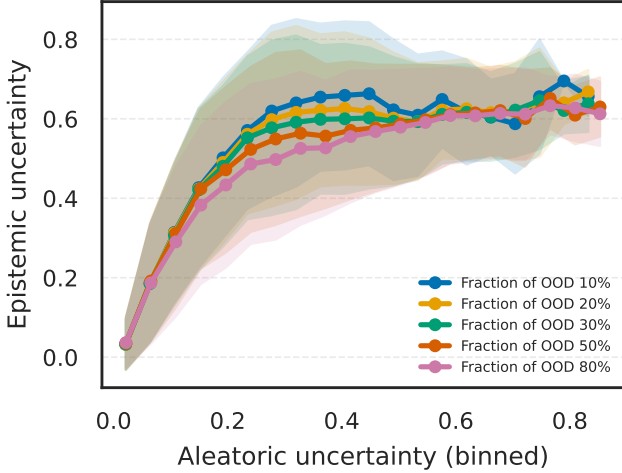

*Figure 2.* Mean epistemic uncertainty for aleatoric uncertainty. The different curves highlight that despite the correlation stays the same, the values of epistemic uncertainty decrease for larger proportions of OOD images within the test set, which is contrary to popular belief.

The boundedness of entropy exposes two fundamental failure modes. First, so-called *unknown unknowns* may occur when all components extrapolate confidently yet incorrectly, yielding low entropy despite being OOD. Second, even under maximal disagreement, the additional (epistemic) uncertainty that can be expressed beyond the upper ensemble envelope is limited by $\log M$. As a result, different degrees of distributional shift may map to similar uncertainty values, causing epistemic signals to saturate. Predictive uncertainty can therefore only express ignorance within the representational envelope spanned by the ensemble components and their weighting mechanism: it quantifies disagreement among existing hypotheses, not ignorance about missing ones. Even under full diversity of hypothesis (de Mathelin et al., 2025) this is a fundamental limitation mathematically enforced by the entropy bound.

**Empirical coupling between aleatoric and epistemic uncertainty.** Figure 1 showed that the epistemic component of uncertainty grows as aleatoric does too. As a consequence of this behavior we plot in Figure 2 the mean epistemic uncertainty as a function of aleatoric uncertainty for this experiment. Contrary to their common treatment as orthogonal components, epistemic uncertainty increases monotonically with aleatoric uncertainty, exhibiting an approximately logarithmic relationship. This trend mirrors findings reported

in recent large-scale benchmarks (Mucsányi et al., 2024), where epistemic and aleatoric components were likewise observed to be coupled, and is also in line with the fact that the orthogonality criteria for clean disentanglement is rarely met (de Jong et al., 2026). Although the predictive entropy admits the additive decomposition in (2), both terms are nonlinear functions of the same predictive mixture (1). In finite ensembles, this nonlinearity can induce empirical coupling between estimated components.

This empirical dependence undermines the common intuition that epistemic uncertainty can be cleanly isolated as a measure of model ignorance. In practice, variations in the input $x$ that increase output noise (aleatoric uncertainty) also tend to induce increased disagreement across models. As a result, the mutual information term does not reflect purely epistemic quantity, but rather a composite effect entangled with aleatoric variability (see Appendix E.5). The decomposition in Equation (2) may hold algebraically, yet it does not guarantee that the resulting terms correspond to distinct or operationally meaningful sources of uncertainty.

Taken together, these limitations are not only artifacts of finite mixtures, but structural consequences of the nonlinear nature of entropy, regardless of the approximation.[2] As such, they cannot be resolved by improved disentanglement of aleatoric and epistemic terms alone. We conclude that

---

[2]Even in analytically tractable settings like Gaussian Processes, $\mathbb{E}_{p(\theta|D)}[\mathbb{H}[Y|x,\theta,\mathcal{D}]] \leq \mathbb{H}[Y|x,\mathcal{D}] \leq \log K$, i.e. under the standard definitions, this corresponds to $AU \leq TU \leq \log K$, due to concavity of entropy. As demonstrated in Appendix A.

predictive uncertainty, while theoretically well-founded, is incomplete as a safety signal, agreeing with the conclusions in (Kirsch et al., 2021; Berry et al., 2024). Since predictive entropy is bounded for all samples, both ID and OOD points can occupy the same uncertainty range. This overlap arises because the predictive distribution is defined over the label space $\mathcal{Y}$, so its entropy collapses uncertainty about $\mathcal{D}$ into label-space uncertainty. Therefore, high **predictive uncertainty is not enough** to identify a sample as OOD, and reliable detection requires an explicit notion of domain uncertainty—capturing how likely an input $x$ is under the data-generating distribution $p(x|\mathcal{D})$—rather than relying solely on posterior disagreement in label space.

### 3.2. A Layered View: Domain Uncertainty Comes First

The aforementioned observations motivate a fundamental distinction that is often left implicit: **uncertainty about predictions is not equivalent to uncertainty about the input domain**. Classical aleatoric and epistemic uncertainties are both defined *conditional* on a given input $x$, but neither explicitly accounts for whether $x$ itself is well supported by the data-generating process that produced $\mathcal{D}$. That is, predictive uncertainty pertains *in label space*.

Safety-critical tasks such as OOD detection require an additional notion: how *plausible* an input is under the data-generating distribution itself. We refer to *domain uncertainty* as the uncertainty over the input itself with respect to the training distribution, formalized simply as

$$p(x|\mathcal{D}) \tag{7}$$

Separating domain uncertainty from aleatoric and epistemic therefore addresses a structural blind spot of predictive uncertainty. While the latter quantifies ambiguity given that the input is assumed valid, domain uncertainty explicitly models the validity of this assumption. This separation is particularly critical in safety-relevant settings, where confident predictions on unsupported inputs can lead to systematic and silent failures (Nguyen et al., 2015).

Modeling the input distribution $p(x|\mathcal{D})$ has been proposed as a principled way to detect distributional shifts and assess whether a given input lies within the domain where a model can be expected to perform reliably (Bishop, 1993). This intuition has been revisited in modern deep learning through likelihood-based OOD detection methods. Nalisnick et al. (2019b) show that deep generative models can assign higher likelihoods to OOD data than to ID samples, revealing fundamental difficulties in accurately modeling the true input distribution. Serrà et al. (2020) shed light on this behavior by finding a negative correlation between image complexity estimates and the likelihood assigned by generative models. That work also proposes to leverage complexity estimates to improve OOD detection through scores that combine likeli-

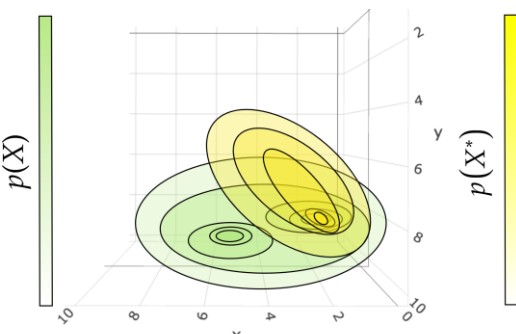

*Figure 3.* Representation of the mismatch between the distribution over inputs at training and test time, $p(X)$ and $p(X^*)$ respectively, which encodes the *domain* uncertainty (Brando et al., 2023).

hood and complexity. This limitation is further examined by Jazbec et al. (2025), who evaluate likelihood in a learned feature space, thereby better aligning density estimation with semantic structure. Other approaches like (Choi et al., 2018) propose ensembling generative models, explicitly framing the OOD detection as the question of "whether an input $x$ should be fed into $p(y|x)$ at all". Despite this refinement, they observe that likelihood estimation remains unreliable for certain tasks and data modalities.

Subsequent work has further studied this limitations. Zhang et al. (2021) systematically investigate why likelihood-based generative methods fail as OOD detectors, showing that high likelihood does not necessarily correspond to semantic similarity to the training distribution. Ren et al. (2019) offer an explanation for these failures by introducing likelihood ratios with respect to a background model, demonstrating that raw likelihood values conflate information about low-level statistics with meaningful distributional membership.

Ideally, we want to model the domain distribution $p(x|\mathcal{D})$ so that it offers a principled quantitative measure of how close a test sample is to the training domain, which is valuable both for binary OOD tasks and for continuous distributional drift such as covariate shift. In (Winkens et al., 2020) authors use a Confusion Log Probability (CLP) metric to distinguish near-OOD from far-OOD, suggesting that how far $p(x|\mathcal{D})$ deviates from the nominal density encapsulates a continuum of distributional shift difficulty. This perspective could be naturally extended to covariate shift: as the marginal $p(x|\mathcal{D})$ shifts gradually away from the training distribution, likelihood values could provide a continuous indication of how strong the shift is and how likely the model is to generalize well on that input.

The nested contours in Figure 3 make this explicit: even samples within the support of the referred $p(X)$ can oc-

cupy more central or more peripheral regions, so being in-distribution does not imply a zero or uninformative domain-level signal. In other words, a well-estimated $p(x|\mathcal{D})$ can serve not only as a binary signal of outliers (as regular OOD methods suggest (Sun et al., 2021)) but also as a predictor of model reliability under both subtle (near) and severe (far) distribution drift.

We argue that uncertainty in machine learning should be decomposed along three axes: aleatoric, epistemic, and domain uncertainty. While the first two quantify uncertainty of the model's predictions, the domain term captures uncertainty about the input's membership in the training domain. The bounded nature of predictive uncertainty makes this separation necessary: without an explicit model of $p(x|\mathcal{D})$, certain out-of-distribution inputs remain fundamentally un-detectable. Therefore, **a full uncertainty representation requires a joint distributional view** of predictive and do-main uncertainty, where $p(y|x,\mathcal{D})$ and $p(x|\mathcal{D})$ are modeled as complementary components. Recognizing domain uncertainty as a distinct quantity clarifies empirical contradictions in OOD detection and motivates evaluation protocols that disentangle predictive ambiguity from distributional nov-elty.

### 3.3. From Parts to Whole: A Joint View of Uncertainty

Let $(x, y)$ denote a new test example. The conditional joint distribution of this pair given the observed dataset $\mathcal{D}$ (data domain) can be factorized as

$$p(x, y|\mathcal{D}) = p(x|\mathcal{D})p(y|x, \mathcal{D}) \tag{8}$$

which follows from the standard rules of joint probability factorization (Brando, 2022).The right hand side term is indeed Equation (1), which contains information about the aleatoric and epistemic uncertainties associated with the conditional prediction. In contrast, the joint distribution together includes the likelihood of observing the input $x$ under the data distribution induced by $\mathcal{D}$. Thus, this joint probability can be interpreted as knowing how likely each input-label pair is in the population, rather than only how likely each label is given an input. Early work by Bishop (1993) suggested multiplying the unconditional to the condi-tional distribution, underscoring the interplay between input and label uncertainty.

*Hybrid models* provide a concrete instantiation of this per-spective (Nalisnick et al., 2019a). For example, hybrid archi-tectures jointly parametrize both $p(x|\mathcal{D})$ and $p(y|x, \mathcal{D})$ in a single framework by combining a deep invertible transfor-mation (such as GLOW (Kingma & Dhariwal, 2018)) with a discriminative predictive model (Nalisnick et al., 2019a). That work finds that jointly training a generative density estimator over inputs together with a discriminative classi-fier improves predictive calibration and robustness, while

preserving state-of-the-art classification accuracy. Crucially, the generative component acts as a regularizer that biases the model toward representations consistent with the data distribution. The work of (Zhang et al., 2020) on open set recognition also adopts a hybrid structure by jointly learning a representation that simultaneously supports classification and input likelihood estimation, yielding improved detec-tion of novel inputs. These works showcase the promising performance of modeling the joint distribution, encouraging learning with this objective.

To assess the potential benefit of an ideal estimator of $p(x|\mathcal{D})$ for our ensemble, we complement the experimen-tal results presented in this paper with Figure 4. The fig-ure reports Risk-Coverage (RC) curves comparing different proxies for predictive uncertainty when ranking challeng-ing samples drawn from a mixture of in-distribution and out-of-distribution data. We observe that ordering sam-ples using epistemic uncertainty yields the weakest results. Using aleatoric uncertainty or maximum softmax proba-bility $p(\hat{y}|x)$ (Hendrycks & Gimpel, 2017) leads to im-proved performance, although a noticeable gap remains. As an upper bound, we include the true joint probability $p(x)p(y|x)$, which is not available at inference time and therefore serves only as an oracle baseline. Importantly, we find that weighting the predictive distribution with the true input density—i.e., incorporating the joint distribu-tion—substantially improves the area under the RC curve, highlighting the potential gains achievable through accurate modeling of $p(x|\mathcal{D})$.

If one aims to model the distribution in Equation (7) directly, they require to integrate over the latent parameters $\phi$ gov-erning the data-generating process, yielding the predictive distribution

$$p(x|\mathcal{D}) = \int_{\Phi} p(\phi|\mathcal{D})p(x|\phi, \mathcal{D})d\phi \tag{9}$$

which mirrors the Bayesian treatment of predictive uncer-tainty in supervised learning. This formulation makes ex-plicit that uncertainty in the input density arises from pa-rameter uncertainty over the dataset $\mathcal{D}$, rather than from observation noise alone.

Inserting this into the conditional expression from Equa-tion (1) gives the expanded expression:

$$p(x, y|\mathcal{D}) = \int_{\Phi} p(\phi|\mathcal{D})p(x|\phi, \mathcal{D})d\phi$$
$$\int_{\Theta} p(\theta|x, \mathcal{D})p(y|x, \theta, \mathcal{D})d\theta \tag{10}$$

where the first integral captures uncertainty associated with the input distribution, while the second corresponds to uncer-tainty in the conditional predictive model. This factorization

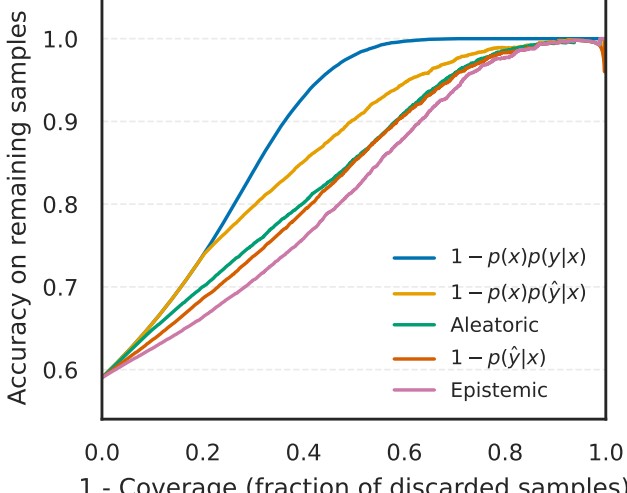

*Figure 4.* Risk-Coverage curves for different notions of uncertainty. Adding the domain term improves the MSP. Test images consisting on 80% ID and 20% OOD are removed according to an uncertainty score and accuracy on the remaining samples is computed. Against popular belief, we see that for this OOD task AU achieves better result than EU.

clarifies that both sources of uncertainty contribute independently to the final predictive belief, and that neglecting either term amounts to an implicit modeling assumption.

Taken together, these lines of research indicate that **a comprehensive treatment of uncertainty should account for all three sources**: the two encoded in the conditional distribution and the domain, rather than conditioning solely on predictive models.

## 4. Alternative Views

In this section we present some alternative views that could contradict our position. These are possible arguments to go against the implementation of a joint distribution as measure of uncertainty.

### 4.1. Modeling $p(x|\mathcal{D})$ is often Too Difficult

A natural but ambitious goal in uncertainty quantification and distributional robustness is to explicitly model the input distribution $p(x|\mathcal{D})$, since having access to this density would, in principle, allow for principled uncertainty estimates and out-of-distribution detection. However, learning accurate high-dimensional density models remains a very challenging task in practice (Nalisnick et al., 2019b). Even state-of-the-art deep generative models such as normalizing flows, VAEs, and GANs only approximate the true data distribution poorly in many realistic settings, and these ap-

proximations often degrade, especially in low-density or sparsely sampled regions of the input space. As with previous observations in generative modeling (Serrà et al., 2020; Jazbec et al., 2025), the "curse of dimensionality," complex multimodal structures, and insufficient inductive biases make exact recovery of $p(x|\mathcal{D})$ infeasible for complex data domains (e.g., images, text, multimodal data) under standard training regimes.

Because of this difficulty, a large body of practical work avoids directly estimating $p(x|\mathcal{D})$ altogether and instead relies on alternative mechanisms that do not require explicit density estimation. For example, some effective OOD detection methods use discriminative or feature-based scores rather than likelihood models of $x$, such as distance or feature-space scoring methods (Lu et al., 2024; Sun et al., 2022) or, even, conformal prediction techniques (Hechtlinger et al., 2019). In practice, this extra cost can be reduced, e.g., by using $p(x|\mathcal{D})$ as a pre-filter before invoking $p(y|x, \mathcal{D})$, as suggested by (Choi et al., 2018), or by using hybrid architectures with shared components. However, the limitation persists even when computational constraints are relaxed, so the core issue is structural rather than computational.

### 4.2. Disentanglement Limitations of the Joint as Uncertainty

Despite the limitations of information-theoretic approaches for disentangling uncertainty sources, the standard decomposition of predictive uncertainty remains conceptually useful. Our proposed comprehensive uncertainty assessment, however, adopts a different perspective: by operating on the joint predictive distribution $p(x, y|\mathcal{D}) = p(x|\mathcal{D})p(y|x, \mathcal{D})$, it captures uncertainty at the level of the data–label pair rather than separating epistemic and aleatoric contributions of the conditional predictor.

A consequence of this formulation is that epistemic and aleatoric uncertainty remain coupled within the predictive distribution. While the entropy of the joint distribution admits a decomposition analogous to that of conditional predictive entropy and obeys similar bounds (cf. Equations (18) and (33)), the joint itself does not provide direct access to the uncertainty sources of the conditional model $p(y|x, \mathcal{D})$ itself. In principle, this joint-space perspective could be extended by ensembling generative models for $p(x|\mathcal{D})$, analogously to ensemble methods used for conditional predictors, as explored in prior work on deep generative models and likelihood-based OOD detection (Nalisnick et al., 2019b; Choi et al., 2018). Such an approach would allow uncertainty to be extrapolated to the unconditional input distribution by treating the modeling of the unconditional in the same way as the conditional.

Nevertheless, relying solely on joint uncertainty conflates

uncertainty about the input domain with uncertainty about the predictive mapping. As a result, it does not reveal whether high uncertainty arises from irreducible data noise, parameter uncertainty in the conditional predictor, or limitations in the representation of the input distribution. Formal derivations supporting these observations are provided in Appendices B and D.

## 5. Conclusion

In this position paper, we revisited the classical information-theoretic decomposition of predictive uncertainty into epistemic and aleatoric components and highlighted fundamental limitations of its practical interpretation. We showed that, for mixture-based predictors such as ensembles, epistemic uncertainty derived from model disagreement (mutual information) is inherently constrained by ensemble size. In particular, we derived entropy bounds that limit the additional uncertainty that can be expressed beyond the best single component.

These bounds help explain why entropy-based uncertainty measures are poorly suited for tasks such as out-of-distribution detection and covariate shift detection, a concern also raised in a recent position work (Li et al., 2025). Due to saturation effects and capacity limits, epistemic uncertainty signals cannot reliably distinguish between in-distribution and out-of-distribution inputs, even under substantial distributional shift.

We argue that meaningful assessment of uncertainty in such settings requires explicit consideration of data plausibility through the data-generating process. Although accurately estimating $p(x|\mathcal{D})$ remains challenging, uncertainty frameworks should aim to approximate this quantity in a graded manner, rather than relying on predictive uncertainty alone. Such estimates should move beyond binary in- or out-of-distribution decisions and instead reflect the continuum of distributional deviations encountered in practice. Our position is that integrating estimates of this into the predictive distribution is necessary to account more fully for uncertainty sources and model failure modes, and we hope this work motivates further research in this direction.

## Acknowledgements

The research leading to these results has received funding from the Horizon Europe Programme under the Horizon Europe Programme under the AI4DEBUNK Project (https://www.ai4debunk.eu), grant agreement num. 101135757. Additionally, this work has been partially supported by the predoctoral grants FI-STEP (2025 STEP 00127) from the Research and University Department of the Generalitat de Catalunya co-funded by the "Fondo Social Europeo Plus".

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

# A. Derivation of Information Theory Equation (2)

Starting with the predictive distribution from Equation (1) and assuming that it is a classification problem with $K$ classes, we express its entropy:

$$\mathbb{H}[Y|x, \mathcal{D}] = -\mathbb{E}_{p(y|x,\mathcal{D})}[\log p(y|x, \mathcal{D})] = -\sum_y p(y|x, \mathcal{D}) \log p(y|x, \mathcal{D}) \leq \log K \tag{11}$$

Now we express the conditional entropy given $\theta$:

$$\mathbb{H}[Y|x, \theta, \mathcal{D}] = -\sum_y p(y|x, \theta, \mathcal{D}) \log p(y|x, \theta, \mathcal{D}) \tag{12}$$

And now the expected conditional entropy:

$$\mathbb{E}_{p(\theta|\mathcal{D})}\big[\mathbb{H}[Y|x, \theta, \mathcal{D}]\big] = \int p(\theta|x, \mathcal{D})\mathbb{H}[Y|x, \theta, \mathcal{D}]d\theta \tag{13}$$

Now we work with the following entropy identity:

$$\mathbb{H}(A|B, C) = \mathbb{H}(A|C) - \mathbb{I}(A; B|C) \tag{14}$$

So, having $A = Y$, $B = \theta$ and $C = (x, \mathcal{D})$ gives:

$$\mathbb{H}[Y|x, \theta, \mathcal{D}] = \mathbb{H}[Y|x, \mathcal{D}] - \mathbb{I}[Y; \theta|x, \mathcal{D}] \tag{15}$$

Since, by definition, the conditional entropy with a random variable $\theta$ is the expectation of that entropy, that is, $\mathbb{H}[y|x, \theta, \mathcal{D}] = \mathbb{E}_{p(\theta|\mathcal{D})}\big[\mathbb{H}[y|x, \theta, \mathcal{D}]\big]$, we can rearrange Equation (15) to finally obtain[3]:

$$\mathbb{H}[Y|x, \mathcal{D}] = \mathbb{E}_{p(\theta|\mathcal{D})}\big[\mathbb{H}[Y|x, \theta, \mathcal{D}]\big] + \mathbb{I}[Y; \theta|x, \mathcal{D}] = \mathbb{E}\big[\mathbb{H}[Y|x, \mathcal{D}]\big] + \mathbb{I}[Y; \theta|x, \mathcal{D}] \tag{16}$$

# B. Proof of Joint Entropy

We start with the distribution we are modeling, which is the join in Equation (8):

$$p(x, y|\mathcal{D}) = p(x|\mathcal{D})p(y|x, \mathcal{D}) \tag{17}$$

Now, given that $x$ and $y$ are realization of the random variables $X \in \mathcal{X}$ and $Y \in \mathcal{Y}$, we can compute the entropy of the random variables:

$$\begin{aligned}
\mathbb{H}[X, Y|\mathcal{D}] &= -\sum_x \sum_y p(x, y|\mathcal{D}) \log p(x, y|\mathcal{D}) \\
&= -\sum_x \sum_y p(x|\mathcal{D})p(y|x, \mathcal{D}) \log \big(p(x|\mathcal{D})p(y|x, \mathcal{D})\big) \\
&= -\sum_x \underbrace{\sum_y p(y|x, \mathcal{D})}_{=1} p(x|\mathcal{D}) \log p(x|\mathcal{D}) - \sum_x \sum_y p(x|\mathcal{D})p(y|x, \mathcal{D}) \log p(y|x, \mathcal{D}) \\
&= \mathbb{H}[X|\mathcal{D}] + \sum_x p(x|\mathcal{D}) \underbrace{\Big(-\sum_y p(y|x, \mathcal{D}) \log p(y|x, \mathcal{D})\Big)}_{\mathbb{H}[Y|X=x, \mathcal{D}]} \\
&= \mathbb{H}[X|\mathcal{D}] + \mathbb{E}_{p(x|\mathcal{D})}\mathbb{H}[Y|x, \mathcal{D}] = \mathbb{H}[X|\mathcal{D}] + \mathbb{H}[Y|X, \mathcal{D}]
\end{aligned} \tag{18}$$

---

[3]For two random variables $Y$ and $\Theta$, the conditional entropy is $\mathbb{H}(Y|\Theta) = -\sum_\theta p(\theta) \sum_y p(y|\theta) \log p(y|\theta) = \sum_\theta p(\theta)\mathbb{H}(Y|\Theta = \theta)$.

Alternatively,

$$
\begin{aligned}
\mathbb{H}[Y|X,\mathcal{D}] &= -\sum_{x,y} p(x,y|\mathcal{D}) \log \frac{p(x,y|\mathcal{D})}{p(x|\mathcal{D})} \\
&= -\sum_{x,y} p(x,y|\mathcal{D}) \big( \log p(x,y|\mathcal{D}) - \log p(x|\mathcal{D}) \big) \\
&= -\sum_{x,y} p(x,y|\mathcal{D}) \log p(x,y|\mathcal{D}) + \sum_{x,y} \underbrace{p(x,y|\mathcal{D})}_{p(x|\mathcal{D})p(y|x,\mathcal{D})} \log p(x|\mathcal{D}) \\
&= \mathbb{H}[X,Y|\mathcal{D}] - \mathbb{H}[X|\mathcal{D}]
\end{aligned}
\tag{19}
$$

And rearranging we arrive to the desired expression.

## C. LogSumExp Derivation for Conditional Entropy

The predictive distribution in Equation (1) can be rewritten in log-space as

$$
\log p(y \mid x, \mathcal{D}) = \log \sum_{m=1}^{M} \exp(\log w_m + \log p(y \mid x, \theta_m, \mathcal{D}))
\tag{20}
$$

where $w_m = p(\theta_m|x,\mathcal{D})$. Defining

$$
a_m(y) = \log w_m + \log p(y \mid x, \theta_m, \mathcal{D}),
$$

this expression corresponds to the LogSumExp operator

$$
\mathrm{LSE}(a_1, \ldots, a_M) = \log \sum_{m=1}^{M} \exp(a_m)
\tag{21}
$$

The LogSumExp satisfies the standard bounds

$$
\max_m \{a_1, \ldots, a_M\} \leq \mathrm{LSE}(a_1, \ldots, a_M) \leq \max_m \{a_1, \ldots, a_M\} + \log M
\tag{22}
$$

Now, if we take the negative of the expectation over the predictive distribution, since the LSE satisfies the equality in (20) and $\mathbb{H}[Y \mid x, \mathcal{D}] = -\mathbb{E}_{p(y|x,\mathcal{D})}[\log p(y \mid x, \mathcal{D})]$. In fact, the negative entropy is the convex conjugate of the LogSumExp function.

$$
-\mathbb{E}_{p(y|x,\mathcal{D})} \big[ \max_m \{a_1, \ldots, a_M\} \big] \geq \mathbb{H}[Y|x,\mathcal{D}] \geq -\mathbb{E}_{p(y|x,\mathcal{D})} \big[ \max_m \{a_1, \ldots, a_M\} \big] - \log M
\tag{23}
$$

If we unwrap the $a_m$ factors:

$$
-\mathbb{E}_{p(y|x,\mathcal{D})} \big[ \max_m \{\log w_m + \log p(y|x,\theta_m,\mathcal{D})\}_{m=1}^{M} \big] \geq \mathbb{H}[Y|x,\mathcal{D}] \geq -\mathbb{E}_{p(y|x,\mathcal{D})} \big[ \max_m \{\log w_m + \log p(y|x,\theta_m,\mathcal{D})\}_{m=1}^{M} \big] - \log M
$$

If we give equal weight to each of the models, like in a Deep Ensemble, we have $p(\theta_m|x,\mathcal{D}) = \frac{1}{M}$ and the expression reduces to

$$
-\mathbb{E}_{p(y|x,\mathcal{D})} \big[ \max_m \{\log p(y|x,\theta_m,\mathcal{D})\}_{m=1}^{M} \big] + \log M \geq \mathbb{H}[Y|x,\mathcal{D}] \geq -\mathbb{E}_{p(y|x,\mathcal{D})} \big[ \max_m \{\log p(y|x,\theta_m,\mathcal{D})\}_{m=1}^{M} \big]
\tag{24}
$$

Which in the cases where we only have one model $M = 1$ both bounds converge to the negative of the maximum log-likelihood of the single model we have. As we increase the number of models in our ensemble the term $\log M$ in the upper bound increases monotonically, but the term in the expectation can also vary because it is still dependent on the models. Specifically, the following will hold if we add a new model to our ensemble

$$
\max_m \{\log p(y|x,\theta_m,\mathcal{D})\}_{m=1}^{M} \leq \max_m \{\log p(y|x,\theta_m,\mathcal{D})\}_{m=1}^{M+1}
\tag{25}
$$

## D. LogSumExp Bounds for Joint Predictive Models

We consider a joint predictive distribution that factorizes as

$$p(x, y|\mathcal{D}) = p(x|\mathcal{D})\, p(y|x, \mathcal{D}), \tag{26}$$

where both the marginal and conditional distributions are approximated by finite ensembles. Specifically, let

$$p(x|\mathcal{D}) = \int_{\Phi} p(\phi|\mathcal{D})p(x|\phi, \mathcal{D})d\phi \approx \sum_{k=1}^{K} u_k\, p(x|\phi_k, \mathcal{D})$$

$$p(y|x, \mathcal{D}) = \int_{\Theta} p(\theta|x, \mathcal{D})p(y|x, \theta, \mathcal{D}) \approx \sum_{m=1}^{M} w_m\, p(y|x, \theta_m, \mathcal{D}) \tag{27}$$

with $\sum_k u_k = 1$ and $\sum_m w_m = 1$ again.

Define

$$a_k(x) = \log u_k + \log p(x|\phi_k, \mathcal{D}), \qquad b_m(x, y) = \log w_m + \log p(y|x, \theta_m, \mathcal{D}). \tag{28}$$

Then the joint log-density can be written as the sum of two LogSumExp operators:

$$\log p(x, y|\mathcal{D}) = \mathrm{LSE}_K\big(a_1(x), \ldots, a_K(x)\big) + \mathrm{LSE}_M\big(b_1(x, y), \ldots, b_M(x, y)\big). \tag{29}$$

**Lemma D.1** (Sum of LogSumExp Bounds). *For all $(x, y)$, the joint log-density satisfies*

$$\max_k a_k(x) + \max_m b_m(x, y) \;\leq\; \log p(x, y|\mathcal{D}) \;\leq\; \max_k a_k(x) + \max_m b_m(x, y) + \log K + \log M. \tag{30}$$

*Proof.* Applying the standard LogSumExp bound $\max_i v_i \leq \mathrm{LSE}(v) \leq \max_i v_i + \log n$ to each term independently yields

$$\mathrm{LSE}_K(a(x)) \geq \max_k a_k(x), \qquad \mathrm{LSE}_M(b(x, y)) \geq \max_m b_m(x, y)$$

and similarly for the upper bounds. Summing the inequalities gives the result. $\square$

Equivalently, since

$$p(x, y|\mathcal{D}) = \sum_{k=1}^{K} \sum_{m=1}^{M} u_k w_m\, p_k(x)\, p_m(y|x), \tag{31}$$

the joint distribution can be interpreted as a mixture with $KM$ components, and the bound in Lemma D.1 is equivalent to the standard LogSumExp bound with margin $\log(KM)$.

**Corollary D.2** (Joint Entropy Bounds). *The joint entropy satisfies*

$$- \mathbb{E}_{p(x,y|\mathcal{D})}\big[\max_k a_k(X) + \max_m b_m(X, Y)\big] - \log(KM)$$

$$\leq \mathbb{H}[X, Y|\mathcal{D}] \leq \tag{32}$$

$$- \mathbb{E}_{p(x,y|\mathcal{D})}\big[\max_k a_k(X) + \max_m b_m(X, Y)\big]$$

*Proof.* The result follows by taking the negative expectation of the bounds in Lemma D.1 with respect to $p(x, y|\mathcal{D})$ and using linearity of expectation. $\square$

Using the law of total expectation and the supposition of $u_k = \frac{1}{K}$ and $w_m = \frac{1}{M}$ the expression arrives to

$$- \mathbb{E}_{p(x|\mathcal{D})}\big[\max_k \log p(x|\phi_k, \mathcal{D}) + \mathbb{E}_{p(y|x,\mathcal{D})}[\max_m \log p(y|x, \theta_m, \mathcal{D})]\big]$$

$$\leq \mathbb{H}[X, Y|\mathcal{D}] \leq \tag{33}$$

$$- \mathbb{E}_{p(x|\mathcal{D})}\big[\max_k \log p(x|\phi_k, \mathcal{D}) + \mathbb{E}_{p(y|x,\mathcal{D})}[\max_m \log p(y|x, \theta_m, \mathcal{D})]\big] + \log(KM)$$

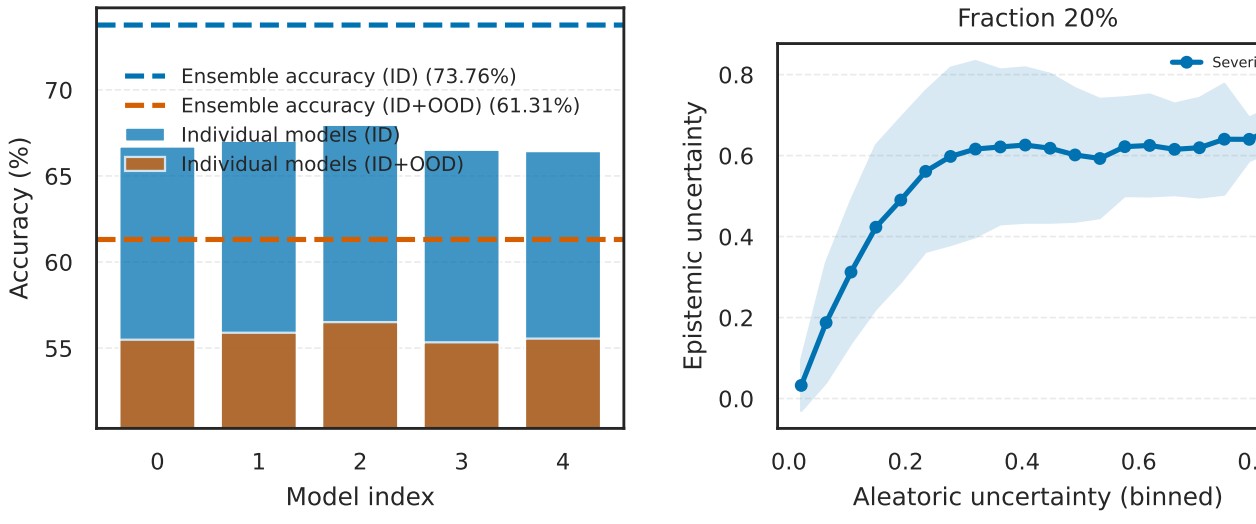

(a) Individual and global accuracies for the models of the ensemble including the OOD points (orange) and not doing so (blue).

(b) EU versus binned AU for the mixed setting (OOD fraction 20%).

*Figure 5.* Mixed test set (80% ID CIFAR-10 + 20% OOD SVHN).

This shows that when both the input distribution and the conditional predictive distribution are modeled as ensembles, the joint entropy behaves as that of a $KM$-component mixture. The logarithmic slack term $\log(KM)$ arises from the nonlinear interaction between model averaging and the logarithm, and cannot be attributed solely to either aleatoric or epistemic uncertainty. Consequently, even at the joint level, total uncertainty cannot in general be decomposed additively into expected conditional entropy and mutual information without ignoring this interaction.

## E. Experimental Results

### E.1. Setup Specifications

All experiments are conducted using simple convolutional neural networks trained on the CIFAR-10 dataset with no transformation applied. Models are trained for 50 epochs using the Adam optimizer (Kingma & Ba, 2015) with a learning rate of $10^{-3}$.

Although early stopping is not employed, the original training set is split into 80% training and 20% validation data. This split is used solely to limit repeated exposure to the same samples during testing, rather than for model selection.

Each experiment is repeated five times with different random seeds. For every run, the individual components of the ensemble are initialized with distinct seeds, which implicitly promotes diversity among ensemble members.

During testing, the same fixed test subset is used across all five runs. This design choice ensures that all reported results are averaged over identical test samples; reshuffling the test set between runs would otherwise introduce unwanted variability and compromise comparability.

### E.2. Accuracy of used models.

The models used to obtain the results shown in Figures 1 and 2 reached accuracies close to 68% in CIFAR10. When evaluated in the mixed test set with 20% of SVHN images (OOD), their accuracies does indeed decrease. In average, this downgrade coincides with a 20% less accuracy when computed in the full mixed dataset. If OOD are not taken into account, the accuracy over the ID samples remains as originally. In addition to the uncertainty capabilities of ensembles, aggregating models also help improve the performance. Figure 5 shows the individual and total accuracies over the full and ID-subset of the test data.

### E.3. Changing SVHN fraction

We can see the dependency between epistemic and aleatoric terms in Figure 2. With increasing fractions of SVHN images the behavior found in Figure 1 is still observed. We repeat those experiments with increasing fractions of SVHN images in the test set of CIFAR10 and plot the results in Figure 6.

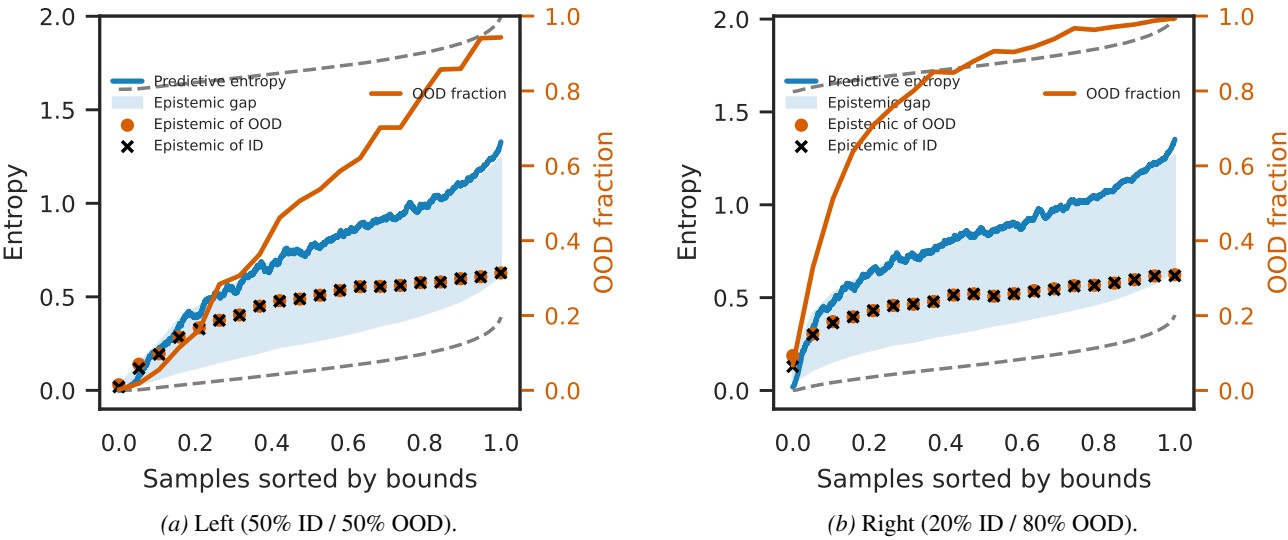

*(a)* Left (50% ID / 50% OOD).

*(b)* Right (20% ID / 80% OOD).

*Figure 6.* Bound plots under ID/OOD mixtures (CIFAR-10 + SVHN).

### E.4. Corrupting test set with CIFAR10-C

We repeat some experiments with a mixed test set of normal CIFAR10 and CIFAR10-C corrupted images with gaussian noise at different severities. In this case, since the intervention is only in the covariate space the detection of corrupted images with Deep Ensembles is even worse. However, the accuracy drop is not as critical as for purely OOD samples like SVHN. These result also make sense with the claims made in (Winkens et al., 2020) about *near-OOD* and *far-OOD* tasks. For CIFAR10, SVHN images are considered far-OOD and CIFAR10-C near-OOD, since the latter are semantically more similar to CIFAR10. For that reason, the percentage of points OOD on the high entropy regions of the plot is higher for SVHN than for CIFAR10-C. It is precisely this semantic similarity which makes the near-OOD detection task more difficult, specially when using uncertainty estimates because we cannot differentiate high entropy due to input ambiguity or unfamiliarity.

In line with this, Figures 7 and 8 further corroborate that epistemic uncertainty alone is not sufficient for OOD analysis in the *near-OOD* regime: ID and corrupted (OOD) samples exhibit very similar epistemic values along the bound-based ordering, yielding only weak separation even as the OOD fraction or corruption severity increases. At the same time, Figures 9 and 10 show that the coupling between epistemic and aleatoric uncertainty is preserved under these covariate corruptions: across severities and mixture proportions, the epistemic–aleatoric relationship remains largely stable, indicating that the EU–AU correlation is robust to both corruption strength and the prevalence of corrupted samples.

### E.5. Isolating aleatoric contamination

To test whether the observed dependence between epistemic and aleatoric uncertainty could be an artifact of domain shift, we inject *intrinsic* noise by swapping an increasing fraction of CIFAR-10 CAT and DOG labels in the training set (0–50%). Figure 11b shows a clear asymmetry: as contamination grows, accuracy decreases while the mean AU increases, whereas EU remains approximately constant (with at most a mild decrease at high noise). Crucially, the *EU–AU coupling* is preserved across contamination levels (Figure 11a): the left panel exhibits an almost invariant monotonic relationship between binned AU and EU, indicating that samples with larger AU also tend to exhibit larger EU irrespective of the injected label noise. Overall, label contamination primarily manifests as higher AU and lower accuracy, but it does not modulate the dependence between EU and AU, suggesting that this correlation is not explained by the noise level and is instead intrinsic to the model's uncertainty structure.

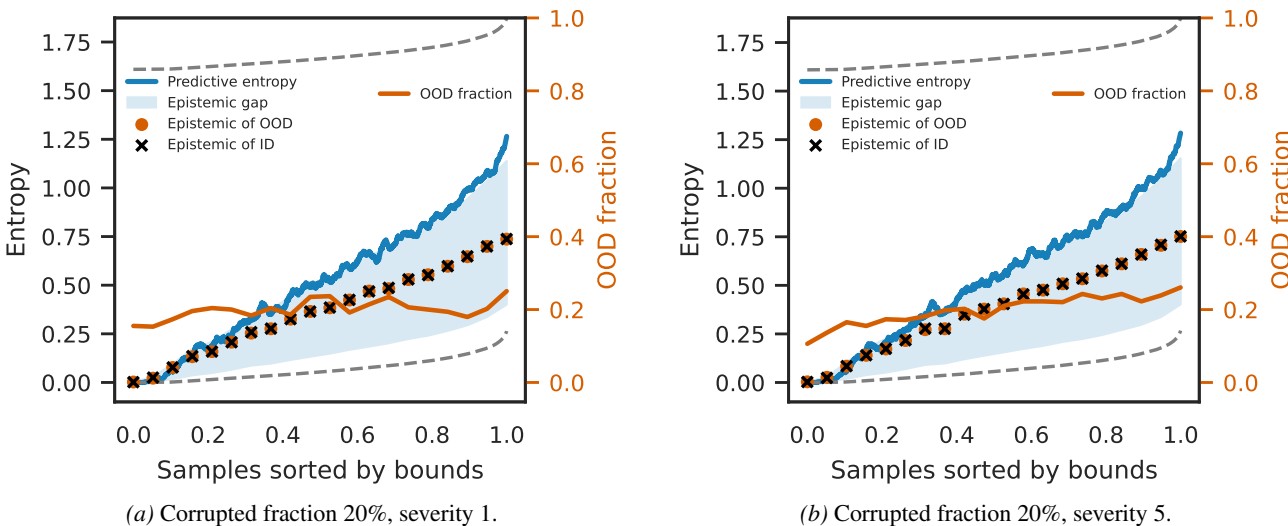

*(a)* Corrupted fraction 20%, severity 1.

*(b)* Corrupted fraction 20%, severity 5.

*Figure 7.* Bound plots for corruption-based covariate shift (CIFAR-10 + CIFAR-10C Gaussian noise).

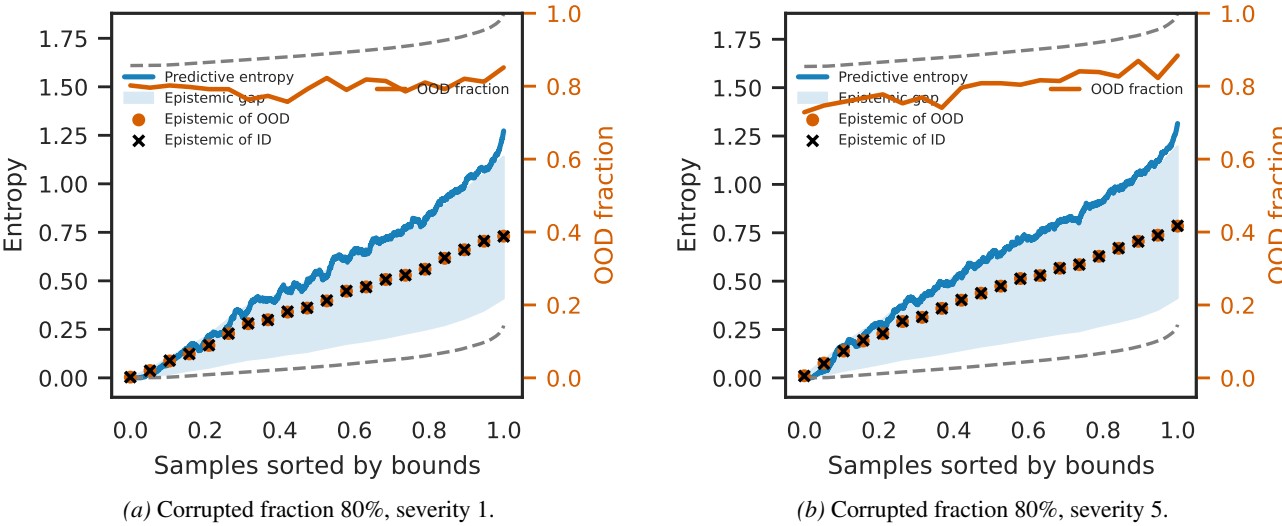

*(a)* Corrupted fraction 80%, severity 1.

*(b)* Corrupted fraction 80%, severity 5.

*Figure 8.* Bound plots for corruption-based covariate shift with high corrupted images prevalence (CIFAR-10 + CIFAR-10C Gaussian noise).

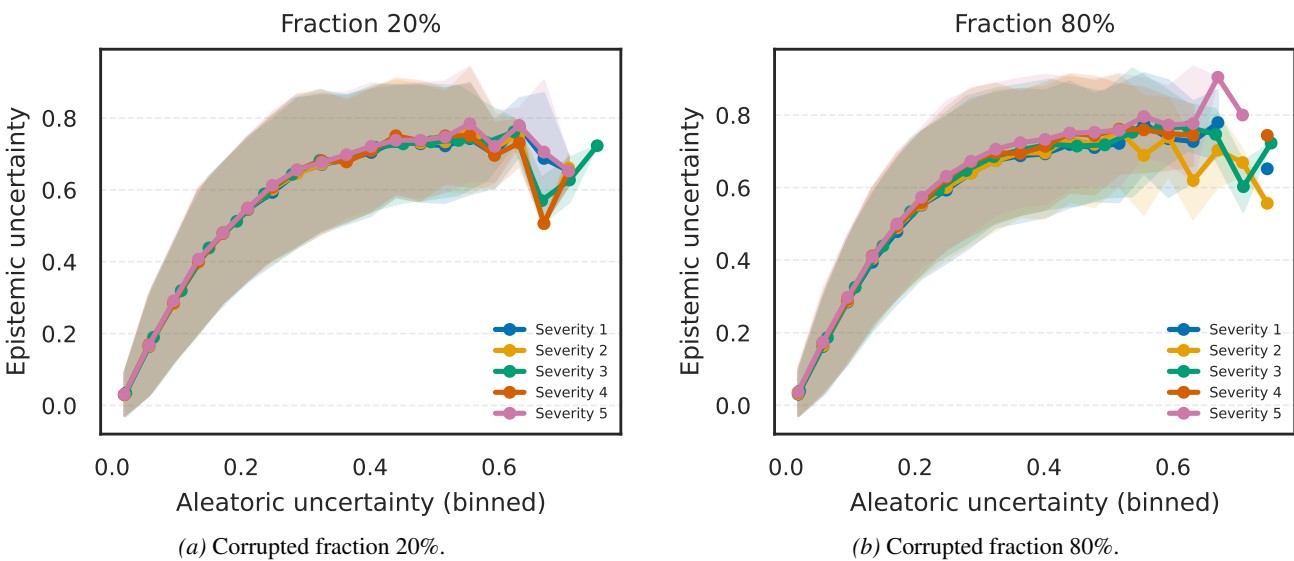

*(a) Corrupted fraction 20%.*   *(b) Corrupted fraction 80%.*

*Figure 9.* EU versus binned AU under corruption-based covariate shift (CIFAR-10C Gaussian noise).

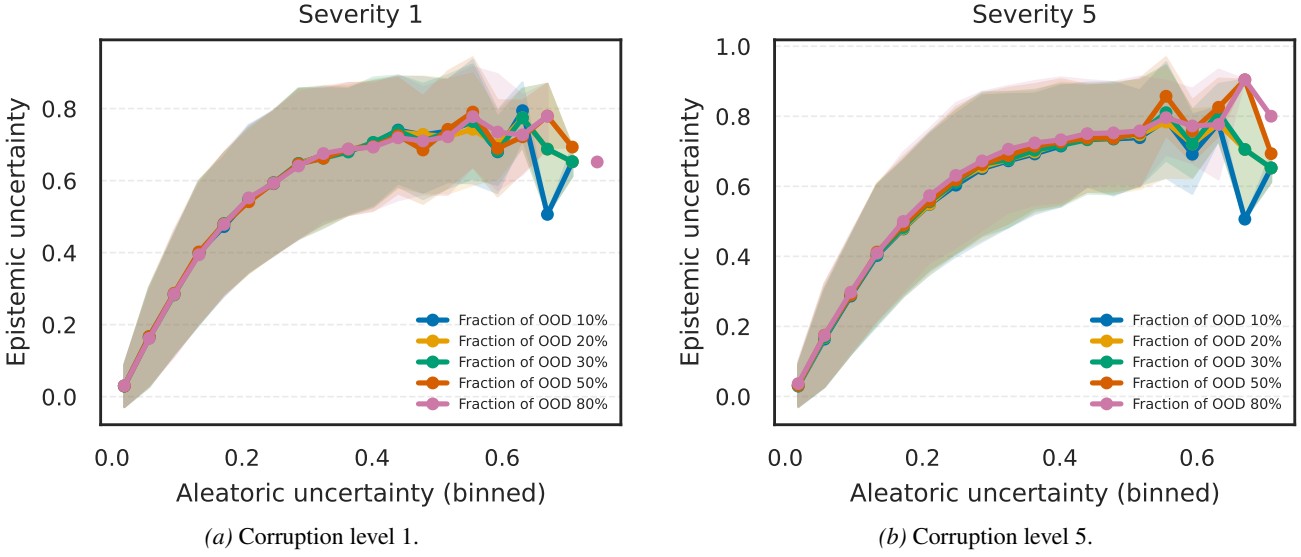

*(a) Corruption level 1.*   *(b) Corruption level 5.*

*Figure 10.* EU versus binned AU for different fractions of corrupted data of CIFAR10C.

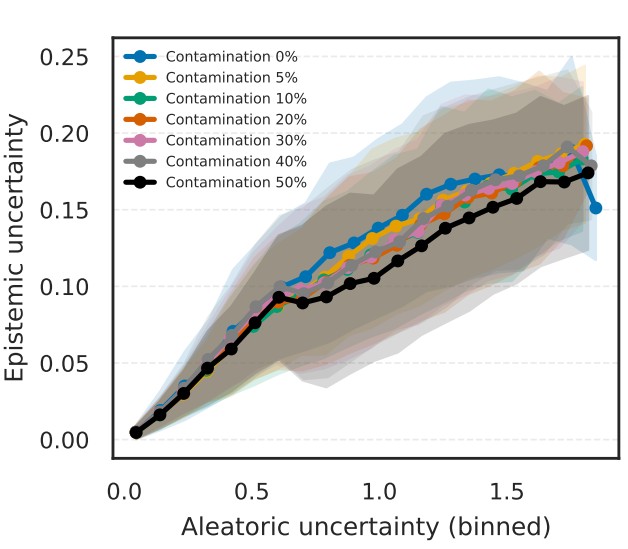

*(a)* EU versus binned AU for different label-contamination rates (0–50%).

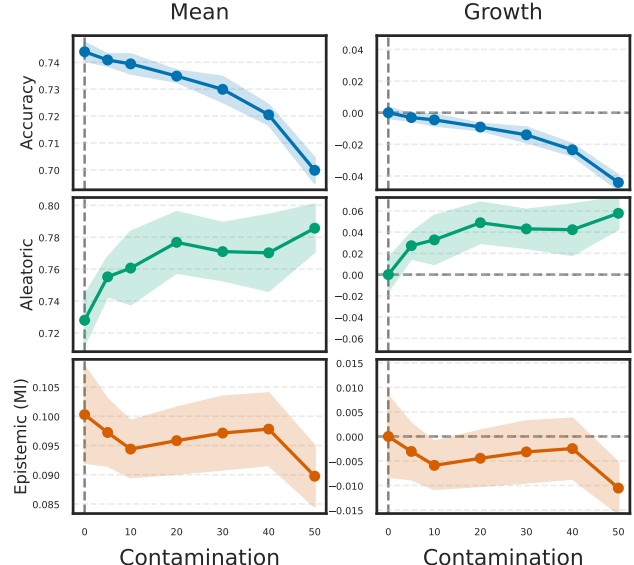

*(b)* Mean (left column) and change with contamination (right column; "Growth") of accuracy, AU, and EU as label contamination increases.

*Figure 11.* Results for contaminated labels

