# OpenReview forum: "Position: Predictive Uncertainty Is Not Enough — Joint Distribution for Full Uncertainty Representation"
_ICML.cc/2026/Position_Paper_Track — ICML 2026 Position Paper Track regular_

### Official Review · Reviewer_wo2H · 2026-03-11

**Significance:** 2
**Argument Clarity:** 2
**Rating:** 4
**Confidence:** 4

**Questions:**

This is somewhat of an opinionated/subjective take so feel free to disregard it (it didn't affect my assessment of the paper, I would just be curious to hear authors's perspective on this): one thing I always found weird about a lot of UQ works that talk about aleatoric vs epistemic uncertainty is that as a motivation they would often use some "safety-critical" application (e.g., autonomous driving) or cite some EU regulation, but then when one talks to practitioners they do not seem to care at all about this decomposition. The only use case where I've seen epistemic uncertainty being used is active learning. So I wonder to what extent is this decomposition a theoretical exercise we've constructed for ourselves as a UQ field, while the outside world has moved on and cares about quite different problems (ie., is the predictive uncertainty reflective of model correctness, hallucination detection in LLMs, selective prediction, adaptive compute via uncertainty etc.). I also wonder to what extent this helped conformal prediction have a bigger adoption among practitioners (compared to some second-order methods like BNNs) since it didn't get lost in this epistemic vs aleatoric exercise

**Alternative Views Section:**

Yes

**Compliance With Llm Reviewing Policy A Conservative:**

Affirmed.

**Discussion Potential:**

2

**Final Justification:**

I thank the authors for their reply to my rebuttal comment. I maintain my current assessment of weak accept - I think the paper is interesting because of its central idea (using domain uncertainty), but for a more positive assessment I still think the experiments would need to be made stronger (even though this is a position paper track)

**Paper Summary:**

The submitted manuscript argues for the position that the uncertainty decomposition into aleatoric and epistemic uncertainty is flawed/underspecified (echoing some of the recent work that has highlighted the same issues). To get around this, the authors propose to move away from predictive uncertainty to joint uncertainty that also incorporates domain uncertainty p(x* | D) (ie., how "OOD" is a given test sample x*).

**Position:**

Yes

**Position In Title:**

Yes

**Related Work:**

3

**Strengths And Weaknesses:**

Strengths:
- I like the main idea of including domain uncertainty p(x|D) into the predictive uncertainty and think that this position paper might serve as a good starting point for more discussions on this topic
- While I do not keep in touch with UQ literature much, I feel like the paper does a good job discussing the related work including the recent papers that also criticise the traditional aleatoric vs epistemic decomposition

Weaknesses:
- I do not understand Figure 1, can you elaborate what it is exactly that you plot there? You sort the examples w.r.t. predictive entropy of a deep ensemble? What do you mean by epistemic gap?
- While it's true that in Nalisnick et al [1] it is shown that p(x|D) might not be useful for OOD, I feel like the authors overlooked some important follow-up work [2] that shows that's mainly an artefact of high-dimensional likelihoods being too sensitive to the input's complexity. See also "semantic likelihood" in [3]
- Even though this is a position paper, I feel like the paper would benefit greatly by having more experiments. The only experiments seem to be on CIFAR/SVHN? It feels a bit of a stretch to cite EU regulation and then only do CIFAR, especially in the age of LLMs/agents? I find the result in Figure 4 interesting, but have hard time trusting it fully since it's CIFAR so would be interesting if it generalises to ImageNet at least or even better to some multiple-choice QA with LLMs
- While you briefly touch upon the computational aspect, I think the manuscript would benefit from a larger discussion on this. Because for predictive uncertainty we only have to train the discriminative model p( y | x, D), but for joint uncertainty we additionally have to train the generative model p(x | D)

[1] Nalisnick et al. Do Deep Generative Models Know What They Don't Know?

[2] Serrà et al. COMPLEXITY AND OUT-OF-DISTRIBUTION DETECTION WITH LIKELIHOOD-BASED GENERATIVE MODELS

[3] Jazbec et al. Generative Uncertainty in Diffusion Models

**Support:**

2

---

> ### Author Rebuttal · Authors · 2026-03-30
>
> We thank the reviewer for the comments and respond below.
>
> - W1: Fig.1 shows how predictive entropy behaves on mixed ID/OOD data relative to the bounds in Eq.5. We evaluate a Deep Ensemble on a mixed test set and, for each sample x, compute predictive entropy, the lower and upper bounds, and the aleatoric/epistemic decomposition from Eq.2. Examples are sorted by lower bound as pointed in Fig.1 caption, we will make it bold. It shows predictive entropy, the two bounds (gray dashed), and a shaded region TU−AU, i.e. “epistemic gap”: it visualizes EU since its lower edge is AU and its height is EU, this will be clarified in the caption. Orange dots (OOD) and black crosses (ID) show epistemic values, while the orange line (right axis) shows the OOD fraction along the sorted samples. The key result is that OOD samples are enriched at high entropy, but ID and OOD still overlap strongly in EU and AU, supporting our claim that TU alone is insufficient for reliable OOD detection.
>
> - W2: We agree that [2, 3] are important references and will include it, as they clarify failure modes of likelihood-based generative models. We will apply the following changes:
>   - Line 298 (left): … distribution. [2] shed light on this behavior by finding a negative correlation between image complexity estimates and the likelihood assigned by generative models. That work also proposes to leverage complexity estimates to improve OOD detection through scores that combine likelihood and complexity. This limitation is further examined by [3], who evaluate likelihood in a learned feature space, thereby better aligning density estimation with semantic structure. Other approaches like [8]…
>   - Line 393 (left): As with previous observations in generative modeling [2,3]…
>
> - W3: Our experiments support two claims: AU and EU are not uncorrelated, and they do not distinguish ID from OOD. We therefore kept them minimal to isolate these phenomena. This aligns with prior large-scale benchmarks cited throughout the paper [4] that include ImageNet. For LLMs, [5] shows that this dichotomy becomes even blurrier there. We will make this more explicit in the revision:
>   - Line 41 (right): As we show in this paper, this is a fundamental aspect of the mentioned dichotomy that not only is present in classical models, but gets even blurrier in complex models like LLMs [5].
>
>   These conclusions led us to carry out broader technical work intended to be published in the future.
>
> - W4: We agree that modeling p(x|D) adds computational cost beyond a purely discriminative p(y|x,D). However, this cost is not incidental: it is required to access information about domain support that predictive models do not represent. The point is therefore not an optional efficiency/completeness trade-off, but that complete uncertainty modeling requires information absent from p(y|x,D) alone.
>   - Line 407 (left): In practice, this extra cost can be reduced, e.g., by using p(x|D) as a pre-filter before invoking p(y|x,D), as suggested by [8], or by using hybrid architectures with shared components. However, the limitation persists even when computational constraints are relaxed, so the core issue is structural rather than computational.
>
> - Q1: We agree that EU vs. AU is often not explicitly used by practitioners outside UQ research, although recent initiatives are starting to incorporate it in our experience. This suggests a gap between the theoretical framing of uncertainty and the needs of current applications. We believe this gap will shrink since, due to increasing regulations, projects with higher TRL are starting to be more aware of the need for trustworthiness in AI. The rise of LLMs makes this even more important: recent work [5] calls for reassessing this decomposition in LLMs, [6] shows that classical methods no longer hold in language models, and [7] suggests that the original intuition is partly lost.
>
> We hope these clarifications make our position and practical motivation clearer.
>
> [4] Mucsányi et al., "Benchmarking uncertainty disentanglement: Specialized uncertainties for specialized tasks," NeurIPS, 2024.
>
> [5] Kirchhof et al., "Position: Uncertainty quantification needs reassessment for large-language model agents," arXiv:2505.22655, 2025.
>
> [6] Hou et al., "Decomposing uncertainty for large language models through input clarification ensembling," arXiv:2311.08718, 2023.
>
> [7] Amayuelas et al., "Knowledge of knowledge: Exploring known-unknowns uncertainty with large language models," Findings of ACL, 2024.
>
> [8] Choi et al., "WAIC, but why? Generative ensembles for robust anomaly detection," arXiv:1810.01392, 2018.

---

> > ### Author Rebuttal · Reviewer_wo2H · 2026-04-02
> >
> > Thanks for the informative response. While I like the main idea of incorporating domain uncertainty p(x* | D) and believe it could lead to some interesting discussions, I do share some concerns regarding experimental results raised by `98DZ`. I will maintain my score.
> >
> > As a side/minor remark, as mentioned in my reviewer and as commented on by the authors, it would be beneficial for the authors to include some discussion on the "practicality" of EU vs AU. It's important for UQ community to start questioning this in my opinion, and every paper that does so helps.

---

### Official Review · Reviewer_98DZ · 2026-03-11

**Significance:** 2
**Argument Clarity:** 2
**Rating:** 4
**Confidence:** 4

**Questions:**

1) See Weaknesses.

2) "This factorization clarifies that both sources of uncertainty contribute independently to the final predictive belief, and that neglecting either term amounts to an implicit modeling assumption." --
Does it imply that the correlation between "domain uncertainty" and AU / EU is 0?

3) "model uncertainty in the conditional predictor" (line 387, right column). What is "model uncertainty"?

**Alternative Views Section:**

Yes

**Compliance With Llm Reviewing Policy A Conservative:**

Affirmed.

**Discussion Potential:**

3

**Final Justification:**

I would like to thank the authors for their replies.
The rebuttal addressed most of my concerns.

I still feel that not all the arguments are persuasive (like the one about the boundedness of entropy).
I suggest that authors incorporate the changes they promised (clarify arguments, ROC -> PC curve, discuss alternatives). I think this would strengthen the paper.

Given that those changes will be incorporated, the paper seems like an interesting position and is worth discussing in the community. In response, I adjust my score.

**Paper Summary:**

The paper states a position that it is not enough for out-of-distribution detection to consider only discriminative models $p(y \mid x, \theta)$ and the corresponding Bayesian posterior over parameters $p(\theta \mid D)$, but rather one needs to consider a joint distribution $p(x, y \mid D)$.
Authors say that in the literature, there is confusion and conceptual tension in defining the dichotomy between aleatoric and epistemic uncertainties. Authors also demonstrate that total uncertainty, approximated by a Monte Carlo estimate of the predictive distribution's entropy, is bounded, with an upper bound that depends on the number of models in the ensemble.
As a solution to all these issues, the authors propose introducing an additional type of uncertainty, which they call "domain uncertainty". This new type of uncertainty captures and reflects the degree to which the model is, in principle, reliable for the input data.

**Position:**

Yes

**Position In Title:**

Yes

**Related Work:**

2

**Strengths And Weaknesses:**

## Weaknesses:

 In this section, I refer to weaknesses and raise some questions for clarification.

1) The title and the position, as written in the Introduction, imply the generality of the position, namely that uncertainty quantification needs to be extended to three types of uncertainty. However, the actual motivation, as described in the paper, is only linked to a very specific downstream problem, namely out-of-distribution detection. I suggest that authors specify their position accordingly.
2) One of the points raised by the authors and used as the motivation for the third type of uncertainty concerns the observation that, in practice, estimates of AU and EU appear to be correlated. Moreover, in line 255, the authors say "Rather than behaving as orthogonal components ...", suggesting that ideally AU and EU should be statistically uncorrelated.
**I have several questions here.** Why is the correlation between components considered something bad or flawed that needs to be rectified? Why do one expect aleatoric and epistemic uncertainties to be uncorrelated? Moreover, is it true that the new component, which the authors introduced in the paper, is indeed practically uncorrelated with AU and EU estimates? I kindly ask the authors to elaborate on these questions,
3) I am not sure I understand the point with the bounded entropy of Eq. 3. Specifically, why is it a problem? As soon as the measure provides a correct ordering, why is its boundness (and dependency on M) a drawback? Additionally, is it specifically a problem of Monte Carlo approximation with only a few samples? If one uses a Bayesian approximation that allows for a closed-form derivation, e.g., a last-layer Gaussian, or even if it involves Monte Carlo but many samples can be cheaply obtained, can the problem be resolved?
4) I find the results in Figure 1 strange. There are a lot of works (for a few, please see [1,2]) that use ensembles for out-of-distribution detection on CIFAR-10 vs SVHN, and they report quite decent results. I may have misinterpreted Figure 1 (please correct me if so), but the EU values presented in the paper under review are almost indistinguishable across these datasets. Hence, I am worried and sceptical about the results.
5) I find the paper misses a discussion of any alternatives. In lines 242-248, the authors say:
"Predictive uncertainty can therefore only express ignorance within the representational envelope spanned by the ensemble components and their weighting mechanism: it quantifies disagreement among existing hypotheses, not ignorance about missing ones, a fundamental limitation mathematically enforced by the entropy bound."
I agree with this statement. Indeed, what people are doing in the literature on OoD detection is trying to express uncertainty about "missing hypotheses" using a model trained to distinguish "existing hypotheses". In other words, they are using a model trained under a closed-world assumption to detect a new class. Therefore, the problem of OoD detection is ill-posed in the setting that is typically considered. But the approach the authors suggest in the paper does not resolve the issue.
 Specifically, the model is still forced to choose between existing hypotheses. How and why the joint distribution conceptually helps find the missing ones is not clear. I understand that, mechanically, OoD objects ideally should have low density under $p(x \mid D)$. But does it necessarily mean that if the covariate does not belong to any of the training classes, then $p(x \mid D)$ is negligible?

6) There are several alternatives to the authors' suggestion, even more straightforward than the one proposed in the paper. In paper [2], the authors showed that many practical methods of EU are essentially Bayesian approximations of some ground truth values, and these approximations reduce to model disagreement. In the paper [3], it was shown that disagreement of OoD data is not a natural inductive bias of neural networks. However, one can induce it. This would result in high disagreement on OoD data, without the need to train a density model. This, and other possible alternatives (e.g., see [4]), are not discussed.

7) The authors mentioned confusion in the literature about different definitions of AU and EU. Additionally, they introduced a new type of uncertainty. But it seems problematic to me, as the authors never mentioned what they imply for AU and EU. If AU is reducible, and EU is irreducible, then it is not possible to introduce something else without intersections, as these two notions already cover everything. So, can the authors clarify their definitions of AU and EU?

### Minor:

- Equation (3) is currently written inaccurately. In line 096, $p(\theta \mid D)$ was introduced as a posterior distribution over parameters. Then, in Equation (3) $p(\theta_m \mid D)$ has another meaning.

- line 228 "bot" -> both

- line 379 "fomrmulation" -> formulation

----

[1] Lakshminarayanan, B., Pritzel, A., & Blundell, C. (2017). Simple and scalable predictive uncertainty estimation using deep ensembles. Advances in neural information processing systems, 30.

[2] Kotelevskii, N., Kondratyev, V., Takác, M., Moulines, É., & Panov, M. From risk to uncertainty: Generating predictive uncertainty measures via bayesian estimation. ICLR 2025.

 [3] de Mathelin, A., Deheeger, F., Mougeot, M., & Vayatis, N. (2023). Maximum weight entropy. arXiv preprint arXiv:2309.15704.

 [4] Hechtlinger, Y., Póczos, B., & Wasserman, L. (2018). Cautious deep learning. arXiv preprint arXiv:1805.09460.

**Support:**

2

---

> ### Author Rebuttal · Authors · 2026-03-30
>
> We thank the careful review and address the points below.
>
> - W1: We claim that AU/EU are not uncorrelated and that they miss information about input validity. That is why we state that “any meaningful uncertainty analysis must explicitly account for three sources of uncertainty” and propose the joint dist.. For that, OOD detection serves as a convenient use case to support our results. However, they are not exclusive to it as seen in the correlation of Fig.2; or App.E5 where we do not include OOD samples.
> - W2: We did not intend to claim that AU and EU should ideally be statistically uncorrelated, nor that their observed correlation is itself a flaw. Our point is precisely that, although they are often presented as independent, their estimators behave in correlated ways, as also reported by prior work and own results. We will revise line 255 to avoid suggesting otherwise, e.g. “contrary to their common treatment as orthogonal components...”. Likewise, we do not claim that DU must be independent of AU/EU. Since p(x|D) and p(y|x,D) are evaluated on the same x, some correlation may naturally arise. The point is not to restore an orthogonal tripartition, but independence of DU in Eq.8 makes explicit its different objective, without implying zero correlation. Same for Q2.
> - W3: We agree that boundedness itself should not be presented as a drawback, we will revise the wording accordingly. Our claim is narrower: bounded predictive entropy reveals a structural saturation as an OOD signal, because ID and OOD samples can occupy the same uncertainty range and high-uncertainty points need not be OOD. The issue is therefore not boundedness per se, but the saturation/overlap it permits when uncertainty is used to infer domain atypicality. This is not a MonteCarlo artefact. Even with closed-form Bayesian approximations, predictive entropy remains bounded by log K in K-class classification, and the standard decomposition into expected conditional entropy plus mutual information still holds (cf. W1-W2 for pPjL).
> - W4: Our reading of Fig.1 is not that ensembles cannot yield useful OOD performance, but that good aggregate performance does not imply pointwise OOD identification by EU alone. High-uncertainty regions are enriched in OOD samples, yet ID and OOD still overlap strongly there. This is a prevalence effect rather than a discriminative guarantee. In that sense, Fig.1 helps explain why ensemble-based OOD methods can work reasonably well while still not making EU a clean proxy for domain support. Similar EU tendency across experiments is a consequence of using the same training domain.
> - W6: We agree the manuscript should better discuss alternatives such as [2,3,4]. [2] aligns with our finding that the disentanglement is not enough for OOD (see their App.A) and at the same time relates EU with model disagreement. This disagreement should be optimal under full diversity of hypothesis, i.e, under the correct posterior, as intended to satisfy in [3]. We agree with that for predictive entropy, but we again claim that even under optimal posterior this would not be enough. Additionally, [4] share our intuition that the uncertainty about domain cannot be inferred from p(y|x) and develop a method using conformal prediction on p(x|y). We will apply the following changes:
>   - Line 160: and Appendix A of [2]
>   - Line 247 (left): ones: Even under full diversity of hypothesis [3] this is a fundamental
>   - Line 407 (left): …Sun et al., 2022) or, even, conformal prediction techniques [4].
> - W5: Our claim is not that p(x∣D) finds missing classes or solves open-set recognition, but that it makes explicit a different question: not whether x belongs to one of the training classes, but whether x is supported by the training domain at all; it lies in the input space. Under a closed-world assumption, p(y∣x,D) only quantifies disagreement among existing hypotheses; it lies in the output space and does not express domain support. The joint dist. does not remove that limitation, but tackles both questions. We do not claim that a x without assigned y should have negligible p(x|D) (because input space is not aware of classes) but x whose covariates are strange under D.
> - W7: We will clarify our definitions of AU and EU. We do not assume they form an exhaustive, context-free partition of uncertainty. The reducible/irreducible distinction depends on the task, observed variables, and modeling assumptions. In our paper, AU refers to residual uncertainty over y given x, EU to uncertainty about the modelled x→y relation, and DU to support of x under p(x|D). DU is therefore not introduced as an ad hoc remainder term, but as a quantity made explicit by factorizing the joint dist..
> - Q3: “model” →“epistemic”.
>
> Finally, we will correct Eq.1 by replacing p(θ|D) with q(θ|D) to make an explicit distinction, and fix the typos.
>
> We hope this clarifies the contribution and address the concerns, making the paper a solid and reliable candidate for acceptance.

---

> > ### Author Rebuttal · Reviewer_98DZ · 2026-04-04
> >
> > I would like to thank the authors for their reply.
> >
> > W1. Thank you! Could I kindly ask the authors to elaborate on the results in Figure 2 and the App. E.5? It seems they still include covariate shift (hence are based on a partial case of OoD)?
> >
> > W3. Thank you! Authors say "ID and OOD samples can occupy the same uncertainty range and high-uncertainty points need not be OOD", but doesn't it boil down to the problem of boundedness? If so, then any bounded OoD detection UQ measure is flawed in this sense?
> >
> > W4. It is true that people typically report aggregated (marginal) performance metrics, which do not guarantee per-point discrimination. But can the authors report the marginal ROC-AUC values for their models, to compare them with prior work?
> >
> > W5. Thank you! It is important to discuss explicitly.

---

### Official Review · Reviewer_uTXB · 2026-03-14

**Significance:** 3
**Argument Clarity:** 3
**Rating:** 5
**Confidence:** 3

**Questions:**

I have no specific questions for the authors.

**Alternative Views Section:**

Yes

**Compliance With Llm Reviewing Policy A Conservative:**

Affirmed.

**Discussion Potential:**

4

**Paper Summary:**

The paper argues for the position that Predictive Uncertainty is not enough, and the joint distribution should be considered for full uncertainty representation. It shows that UQ methods that decompose uncertainty into epistemic and aleatoric components are not enough, and shows evidence for their claim empirically.

**Position:**

Yes

**Position In Title:**

Yes

**Related Work:**

3

**Strengths And Weaknesses:**

The paper provides a different perspective on uncertainty quantification. The community has often discussed that the conventional decomposition of uncertainty is not enough, at least when considering them in isolation. The approach to include the joint distribution in the analysis is clever and provides a fresh perspective on the uncertainty quantification. The alternative views suuficiently discusses the modelling difficulty as well as the disentanglement limitation of the Joint distribution. Overall, the idea of considering data plausibility through data generating process can provide good discussion within the UQ community.

**Support:**

3

---

> ### Author Rebuttal · Authors · 2026-03-30
>
> We would like to thank the reviewer for their positive assessment and their accurate summary of the article’s main contribution. The review captures our central argument well: predictive uncertainty alone is not sufficient to fully represent uncertainty, and a more complete description requires considering domain uncertainty alongside the conditional predictive distribution through a joint framework.
>
> The reviewer also correctly identifies two aspects we sought to highlight: firstly, that this perspective offers a new approach to the quantification of uncertainty, and secondly, that the ‘Alternative Viewpoints’ section is important for making explicit both the modelling difficulty and the disentanglement limitations that the joint view still presents.
>
> We are pleased that this review acknowledges the importance of this topic for discussion within the UQ community, and we will use the insights gained from this discussion process to clarify the overall rationale and research focus in the revised manuscript.

---

> > ### Author Rebuttal · Reviewer_uTXB · 2026-04-04
> >
> > Thanks for the rebuttal. I keep my positive evaluation of the work.

---

### Official Review · Reviewer_pPjL · 2026-03-14

**Significance:** 4
**Argument Clarity:** 2
**Rating:** 5
**Confidence:** 4

**Questions:**

1. Generative modelling is discussed briefly in 3.2. But in general, the argument in this is not applicable to generative models, right? Since they do explicitly model domain uncertainty. How do you then make sense of the fact that even in generative models, the likelihood can misidentify OOD regions as ID regions?

2. See weaknesses

**Alternative Views Section:**

Yes

**Compliance With Llm Reviewing Policy A Conservative:**

Affirmed.

**Discussion Potential:**

4

**Final Justification:**

I think its a good paper which is well argued, and authors did a good job with resolving my concerns.

**Paper Summary:**

This paper argues that the traditional aleatoric-epistemic uncertainty decomposition is incomplete. They suggest instead considering the conditional joint distribution of a test pair given the observed dataset. The argument against only considering the predictive posterior distribution is two-fold: 1) Predictive Entropy can extrapolate to low values in OOD regions, leading to confident but incorrect predictions, 2) Epistemic Uncertainty can be bounded above and below by the extremal component in an ensemble and an additional log M term for the upper bound. They also show that epistemic and aleatoric uncertainties are highly correlated. Considering the joint distribution instead contains information about the aleatoric and epistemic uncertainties associated with the conditional prediction, but also contains information about the domain uncertainty (given by the term $p(x|\mathcal{D})$). This paper argues that in an out-of-distribution or distribution shift regime, this additional term is essential.

**Position:**

Yes

**Position In Title:**

Yes

**Related Work:**

3

**Strengths And Weaknesses:**

# Strengths
1. Overall, the position in this paper is intriguing and has high discussion potential.
2. The paper does a good job of outlining some of the problems with predictive entropy, with various citations and has extensively documented such issues.

# Weaknesses
1. The main weakness, in my opinion, is that the argument is a bit loose in some places. Specifically:
  - I am failing to see why the bound in equation 4 exposes the two fundamental failure modes as stated in the paper. I can see that predictive entropy must remain within this bound for both ID and OOD data, but does that necessarily mean that predictive entropy cannot distinguish between ID and OOD samples? It certainly means that a fixed ensemble size predictive entropy is bounded, and that beyond a certain point it cannot distinguish between OOD and "even more" OOD samples, but I am not sure about the first claim. Doesn't Figure 6 contradict the first claim?
  - The argument in the paper, and specifically the bound equation 4, relies on Monte Carlo estimation for Bayesian prediction. I think the paper could do a better job of disambiguating whether the failure modes are a result of this specific approximation or more fundamental. Perhaps a short discussion on Gaussian Process or Linearised Laplace, where prediction is analytically tractable, would have been beneficial.

2. [1] makes a similar argument that is not discussed. It would be interesting to read how their approach fits into the arguments and framework outlined in this paper.


[1] Osband, Ian, et al. "Epistemic neural networks." Advances in Neural Information Processing Systems 36 (2023): 2795-2823.

**Support:**

2

---

> ### Author Rebuttal · Authors · 2026-03-30
>
> We thank the reviewer for the helpful comments and address each point below to clarify our contributions and weaknesses.
>
> - W1: Our claim is not that boundedness alone proves failure, but that it reveals a structural limitation of predictive entropy as an OOD signal. **As predictive entropy is bounded for all samples, both ID and OOD points can occupy the same uncertainty range. Therefore, high predictive entropy does not by itself imply that a sample is OOD.** Eq.4 should be read together with Figs.1,6,7,8, which show this limitation empirically: high-uncertainty regions contain more OOD samples, but uncertain ID samples remain substantially present there. This overlap persists even after disentangling EU and AU, where orange dots and black crosses still occupy similar regions. Fig.6 is thus not a contradiction, but an illustration of the distinction between statistical enrichment and pointwise discrimination. As uncertainty increases, the fraction of OOD samples also increases, so high-entropy regions are enriched in OOD samples. This is a prevalence effect, not a discriminative one: OOD samples become more frequent in those regions, but predictive entropy still does not identify them point-by-point. This is what we meant to convey in lines 219–232, which we will complete with the bolded text above to make it more explicit. Predictive entropy is defined over the label space y, not input atypicality, motivating an explicit domain term to assess OODness in input space directly.
>
> - W2: We use MonteCarlo (MC) because it is a standard approximation of the Bayesian predictive integral (Eq.1). However, the failure modes are not caused by this approximation, but by Bayesian model averaging and entropy itself. To make this distinction explicit, we can consider the continuous (exact) formulation. Defining $\pi(d\theta):=p(\theta|\mathcal{D})d\theta$, we can write $p(y|x,\mathcal{D})=\int p(y|x,\theta,\mathcal{D})\pi(d\theta)$. Concavity of entropy gives $H[Y|x,\mathcal{D}] \geq \int H[Y|x,\theta,\mathcal{D}]\,\pi(d\theta),$ which is the continuous analogue of the discrete mixture result, i.e. $TU \geq AU$, independently of any discretisation. Also, for $K$ classes, predictive entropy remains bounded by $E_{\Theta}[H[Y|x,\Theta,\mathcal{D}]] \leq H[Y|x,\mathcal{D}] \leq \log K.$
>
>   This matters because both properties—the decomposition and the bounded range—already hold in the exact Bayesian formulation. In particular, Eq.2 shows that predictive entropy combines the expected conditional (“AU”) term and a mutual information (“EU”) term regardless of how the posterior is approximated. Likewise, the upper bound $\log K$ is a fundamental property of entropy, not of MC. We will clarify this:
>   - Line 57 (right): … regardless of the approximation:
>
>   This extends naturally to analytically tractable settings such as Gaussian processes or linearised Laplace approximations.Therefore, predictive entropy still marginalises different uncertainty sources into a single bounded scalar. This is precisely our point: the limitations discussed in the paper—its inability, by itself, to separate uncertain ID from OOD samples, and the fact that the quantities commonly labelled “aleatoric” and “epistemic” need not behave as independent components.
>
> - W3: We appreciate the reviewer’s pointer to Epistemic Neural Networks (ENN) and will cite [1]. ENNs identify a limitation aligned with our argument: standard predictors output marginal distributions p(y|x) that collapse different sources of uncertainty into a single distribution. ENNs address this by modelling joint predictions across inputs, whereas our work moves towards the joint distribution p(x,y|D) to separate uncertainty about domain support from uncertainty in label space. Thus, while the goals differ, both works show that marginal predictive distributions are insufficient to capture the structure of uncertainty. We will add this reference in the Sec.2.2.
>   - Line 149: As an alternative, [1] argue that predictions collapse different sources of uncertainty into a single distribution and propose modelling joint predictions across inputs, addressing this loss of information by enriching the output representation.
>
> - Q1: We read the question as whether this three-part uncertainty decomposition extends to generative models. This is interesting as such models already represent the domain. Prior work has studied uncertainty in generative models [2,3]. Eq.9 was intended to open this discussion, since the same factorisation behind Eq.1 also appears when a generative model represents the distribution. However, as the reviewer notes, the variables are reduced: x becomes the target and y disappears. The decomposition therefore reduces to counterparts of AU, tied to the likelihood of x, and EU, tied to the generator of x.
>
> [2] Jazbec et al., "Generative uncertainty in diffusion models," arXiv:2502.20946, 2025.
>
> [3] Berry et al., "Estimating epistemic uncertainty in diffusion models," UAI, 2024.

---

> > ### Author Rebuttal · Reviewer_pPjL · 2026-04-05
> >
> > I am satisfied with the response and intend to keep my positive score,

---

### Decision · Program_Chairs · 2026-04-30

**Decision:**

Accept (regular)

**Comment:**

Based on reviews, rebuttal, and discussion, I'm proposing acceptance for the paper.

All reviewers agreed on the interesting of the position of the paper and the value of the proposed approach. The rebuttal helped in clarifying technical aspects and positioning of the paper.

I would like to encourage the authors to pay particular attention on the following points while preparing the camera ready of the paper:

* The boundedness in Eq.4 was a source of confusion among the reviewers, in particular on the logic implications the authors seem to derive from it to support the position expressed in the section. The authors acknowledge some of the limitations of that statement and, while I don't believe this invalidates the overall point raised in the paper, I encourage the authors to integrate the rebuttal in the final version to avoid any misunderstanding about the intended statement.

* The rebuttal is effective in discussing all the references suggested by the reviewers. Please consider how to smoothly integrate these into the alternative views section while clearly contrasting them with the proposed approach.